# Neural Cover Selection for Image Steganography

**Karl Chahine & Hyeji Kim**
Department of Electrical and Computer Engineering
University of Texas at Austin
Austin, TX 78712
{karlchahine, hyeji.kim}@utexas.edu

## Abstract

In steganography, selecting an optimal cover image—referred to as cover selection—is pivotal for effective message concealment. Traditional methods have typically employed exhaustive searches to identify images that conform to specific perceptual or complexity metrics. However, the relationship between these metrics and the actual message hiding efficacy of an image is unclear, often yielding less-than-ideal steganographic outcomes. Inspired by recent advancements in generative models, we introduce a novel cover selection framework, which involves optimizing within the latent space of pretrained generative models to identify the most suitable cover images, distinguishing itself from traditional exhaustive search methods. Our method shows significant advantages in message recovery and image quality. We also conduct an information-theoretic analysis of the generated cover images, revealing that message hiding predominantly occurs in low-variance pixels, reflecting the waterfilling algorithm's principles in parallel Gaussian channels. Our code can be found at https://github.com/karlchahine/Neural-Cover-Selection-for-Image-Steganography.

## 1 Introduction

Image steganography embeds secret bit strings within typical cover images, making them imperceptible to the naked eye yet retrievable through specific decoding techniques. This method is widely applied in various domains, including digital watermarking (Cox et al. [2007]), copyright certification (Bilal et al. [2014]), e-commerce (Cheddad et al. [2010]), cloud computing (Zhou et al. [2015]), and secure information storage (Srinivasan et al. [2004]).

Traditionally, hiding techniques such as modifying the least significant bits have been effective for embedding small data volumes up to 0.5 bits per pixel (bpp) (Fridrich et al. [2001]). Leveraging advancements in deep learning, recent approaches employ deep encoder-decoder networks to embed and extract up to 6 bpp, demonstrating significant enhancements in capacity (Chen et al. [2022], Baluja [2017], Zhang et al. [2019]). The encoder takes as input a cover image $\mathbf{x}$ and a secret message $\mathbf{m}$, outputting a steganographic image $\mathbf{s}$ that appears visually similar to the original $\mathbf{x}$. The decoder then estimates the message $\hat{\mathbf{m}}$ from $\mathbf{s}$. The setup is illustrated in Fig. 1 (left).

The effectiveness of steganography is significantly influenced by the choice of the cover image $\mathbf{x}$, a process known as cover selection. Different images have varying capacities to conceal data without detectable alterations, making cover selection a critical factor in maintaining the reliability of the steganographic process (Baluja [2017], Yaghmaee and Jamzad [2010]).

From a theoretical standpoint, numerous studies have employed information-theoretic analyses to investigate cover selection and determine the capacity limits of information-hiding systems, thereby identifying the maximum number of bits that can be embedded (Moulin et al. [2000], Cox et al. [1999], Moulin and O'Sullivan [2003]). For instance, in Moulin and O'Sullivan [2003], the steganographic

38th Conference on Neural Information Processing Systems (NeurIPS 2024).

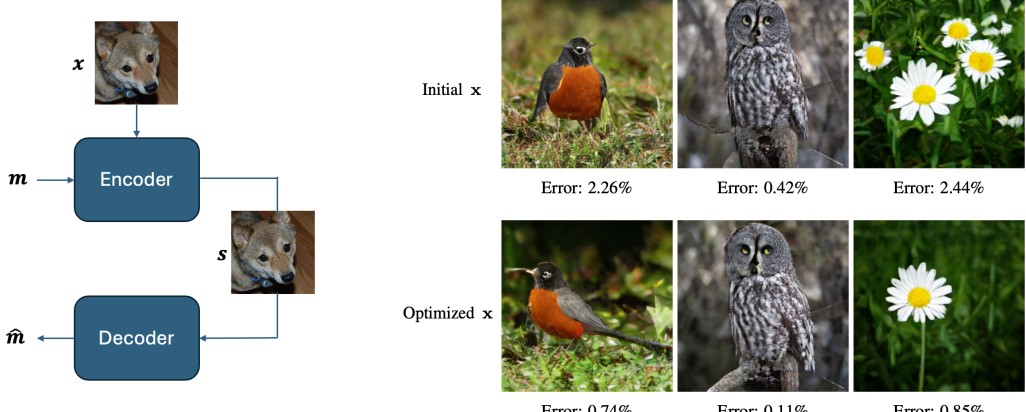

Figure 1: **Left:** Image steganography framework: the encoder takes as input the cover image $\mathbf{x}$ and a secret binary message $\mathbf{m}$ and outputs the steganographic image $\mathbf{s}$. The decoder then estimates $\hat{\mathbf{m}}$ from $\mathbf{s}$. **Right:** Randomly sampled cover images from the ImageNet dataset before and after optimization using our framework (described in Section 3). These optimized images demonstrate a significantly reduced error $||\mathbf{m} - \hat{\mathbf{m}}||$ while maintaining high image quality.

setup is conceptualized as a communication channel where the cover image $\mathbf{x}$ acts as side information. However, such models are based on impractical assumptions: firstly, the steganographic process is additive—where the message $\mathbf{m}$ is simply added to the cover $\mathbf{x}$; and secondly, it presupposes that the cover elements adhere to a Gaussian distribution.

From a practical standpoint, existing techniques for cover selection predominantly rely on exhaustive searches to identify the most suitable cover image. These methods evaluate a variety of image metrics to determine the best candidate from a database. Some strategies include counting modifiable discrete cosine transform (DCT) coefficients to select images with a higher coefficient count for covers (Kharrazi et al. [2006]), assessing visual quality to determine embedding suitability (Evsutin et al. [2018]), and estimating the embedding capacity based on image complexity metrics (Yaghmaee and Jamzad [2010], Wang and Zhang [2019]).

Traditional methods for selecting cover images have three key limitations: (i) They rely on heuristic image metrics that lack a clear connection to steganographic effectiveness, often leading to suboptimal message hiding. (ii) These methods ignore the influence of the encoder-decoder pair on the cover image choice, focusing solely on image quality metrics. (iii) They are restricted to selecting from a fixed set of images, rather than generating one tailored to the steganographic task, limiting their ability to find the most suitable cover.

Recent progress in generative models, such as Generative Adversarial Networks (GANs) (Goodfellow et al. [2020]) and diffusion models (Song et al. [2020], Ho et al. [2020]), have ignited significant interest in the area of guided image generation (Shen et al. [2020], Avrahami et al. [2022], Brooks et al. [2023], Gafni et al. [2022], Kim et al. [2022]). Inspired by these innovations, we propose a novel approach that addresses the aforementioned limitations by treating cover selection as an optimization problem.

In our proposed framework, a cover image $\mathbf{x}$ is first inverted into a latent vector, which is then passed through a pretrained generative model to reconstruct the cover image. This image is processed by a neural steganographic encoder to embed a secret message, followed by a decoder to recover the message. We optimize the latent vector to generate an enhanced cover image $\mathbf{x}^*$, minimizing message recovery errors while preserving the visual and semantic integrity of the image. Fig. 1 (right) presents message recovery errors for randomly selected images before and after optimization. Our approach of optimizing the cover image uncovers a novel way to analyze the transformation from $\mathbf{x}$ to $\mathbf{x}^*$, revealing that the encoder embeds messages in low-variance pixels, analogous to the water-filling algorithm in parallel Gaussian channels. To the best of our knowledge, this is the first work that examines neural steganographic encoders by framing cover selection as a guided image reconstruction problem.

Our contributions are outlined as follows:

1. *Framework.* We describe the limitations of current cover selection methods and introduce a novel, optimization-driven framework that combines pretrained generative models with steganographic encoder-decoder pairs. Our method guides the image generation process by incorporating a message recovery loss, thereby producing cover images that are optimally tailored for specific secret messages (Section 3).

2. *Experiments.* We validate our methodology through comprehensive experimentation on public datasets such as CelebA-HQ, ImageNet, and AFHQ. Our results demonstrate that the error rates of the optimized images are **an order of magnitude lower** than those of the original images under specific conditions. Impressively, this optimization not only reduces error rates but also enhances the overall image quality, as evidenced by established visual quality metrics. We explore this intriguing phenomenon by examining the correlation between image quality metrics and error rates (Section 3.3).

3. *Interpretation.* We investigate the workings of the neural encoder and find it hides messages within low variance pixels, akin to the water-filling algorithm in parallel Gaussian channels. Interestingly, we observe that our cover selection framework increases these low variance spots, thus improving message concealment (Section 4).

4. *Practical considerations.* We extend our guided image generation process to practical applications, demonstrating its robustness against steganalysis and resilience to JPEG compression, as detailed in Section 5.

**Related work.** Recent research has explored the use of generative models in steganography. Zhang et al. [2019] introduced a training framework where steganographic encoders and decoders are trained adversarially, similar to GANs. Yu et al. [2024] harness the image translation capabilities of diffusion models to transform a secret image directly into a steganographic image, bypassing the embedding process, a framework known as coverless steganography (Qin et al. [2019]). Shi et al. [2018] is notably relevant, as they created a GAN framework designed to produce images robust against steganalysis. However, there are three key distinctions: (i) they overlooked message error rates, focusing solely on evading detection, compromising the effectiveness of cover images for message recovery; (ii) they trained their GAN from scratch, failing to leverage the advantages of existing pretrained models; and (iii) the images generated were randomly sampled and not user-selectable, limiting application flexibility.

## 2   Preliminaries

**Image steganography** aims to hide a secret bit string $\mathbf{m} \in \{0,1\}^{H \times W \times B}$ into a cover image $\mathbf{x} \in [0,1]^{H \times W \times 3}$ where the payload $B$ denotes the number of encoded bits per pixel (bpp) and $H, W$ denote the image dimensions. As depicted in Fig. 1 (left), the hiding process is done using a steganographic encoder $Enc$, which takes as input $\mathbf{x}$ and $\mathbf{m}$ and outputs the steganographic image $\mathbf{s}$ which looks visually identical to $\mathbf{x}$. A decoder $Dec$ recovers the message, $\hat{\mathbf{m}} = Dec(\mathbf{s})$ with minimal error rate $\frac{||\mathbf{m}-\hat{\mathbf{m}}||_0}{H \times W \times B}$.

**Cover selection** involves generating the ideal cover image $\mathbf{x}$, to achieve three primary objectives: (i) minimize the error rate as defined above, (ii) ensure that the steganographic image $\mathbf{s}$ visually resembles $\mathbf{x}$ as closely as possible, and (iii) maintain the integrity of the cover image $\mathbf{x}$ using established perceptual quality metrics.

**Denoising Diffusion Implicit Models (DDIMs)** (Song et al. [2020]) are a class of generative models that learn the data distribution by adopting a two-phase mechanism. The forward phase incorporates noise into a clean image, while the backward phase incrementally removes the noise. The formulation for the forward diffusion in DDIM is presented as:

$$\mathbf{x}_t = \sqrt{\alpha_t}\mathbf{x}_{t-1} + \sqrt{1-\alpha_t}\boldsymbol{\epsilon}, \quad \boldsymbol{\epsilon} \sim \mathcal{N}(\mathbf{0}, \mathbf{I}), \tag{1}$$

where $\mathbf{x}_t$ is the noisy image at the $t$-th step, $\alpha_t$ is a predefined variance schedule, and $t$ spans the discrete time steps from $1$ to $T$. The DDIM's backward sampling equation is:

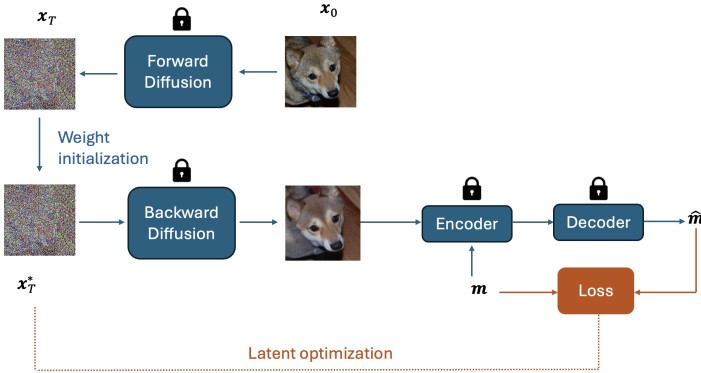

Figure 2: DDIM-based cover selection framework overview. The input cover image $\mathbf{x}_0$ is first converted to the latent space $\mathbf{x}_T$ via forward diffusion. Then, guided the message recovery loss, the latent space is fine-tuned, and the updated cover image is generated via the reverse diffusion process. The DDIM model as well as the steganographic encoder-decoder pair are pretrained.

$$\mathbf{x}_{t-1} = \sqrt{\bar{\alpha}_{t-1}}\mathbf{f}_\theta(\mathbf{x}_t, t) + \sqrt{1 - \bar{\alpha}_{t-1} - \sigma_t^2}\boldsymbol{\epsilon}_\theta(\mathbf{x}_t, t) + \sigma_t^2\boldsymbol{\epsilon}, \quad \mathbf{f}_\theta(\mathbf{x}_t, t) = \frac{\mathbf{x}_t - \sqrt{1 - \bar{\alpha}_t}\boldsymbol{\epsilon}_\theta(\mathbf{x}_t, t)}{\sqrt{\bar{\alpha}_t}},$$

(2)

where $\boldsymbol{\epsilon} \sim \mathcal{N}(\mathbf{0}, \mathbf{I})$, $\bar{\alpha}_t = \prod_{i=1}^{t} \alpha_i$, and $\mathbf{f}_\theta$ is a denoising function reliant on the pretrained noise estimator $\boldsymbol{\epsilon}_\theta$.

This sampling allows the use of different samplers by changing the variance of the noise $\sigma_t$. Especially, by setting this noise to 0, the DDIM backward process becomes deterministic, defined uniquely by the initial variable $\mathbf{x}_T$. This initial value can be seen as a latent code, commonly utilized in DDIM inversion, a process that utilizes DDIM to convert an image to latent noise and subsequently reconstruct it to its original form (Kim et al. [2022]).

**Generative Adversarial Networks (GANs)** (Goodfellow et al. [2020]) are another type of generative model designed to learn the data distribution $p(\mathbf{x})$ of a target dataset through a min-max game between two networks: a generator ($G$) and a discriminator ($D$). The generator creates synthetic samples $G(\mathbf{z})$ from a random noise vector $\mathbf{z}$, drawn from a simple distribution $p(\mathbf{z})$ such as a standard normal. The discriminator evaluates samples it receives—either real data $\mathbf{x}$ from $p(\mathbf{x})$ or fake data from $G$—and tries to accurately classify them as real or fake. The objective of $G$ is to generate data that $D$ mistakes as real, while $D$ aims to distinguish between actual and generated data effectively.

## 3 Methodology

We propose two cover selection methodologies using pretrained Denoising Diffusion Implicit Models (DDIM) and pretrained Generative Adversarial Networks (GAN) (Sections 3.1, 3.2), and compare the performances of the two approaches (Section 3.3). Detailed descriptions of the training procedures are in Appendix B. Broadly speaking, starting with a cover image $\mathbf{x}$ randomly selected from the dataset, we gradually optimize this image to minimize the loss $||\mathbf{m} - \hat{\mathbf{m}}||$. Intriguingly, while our primary focus is on reducing the error rate, we observe that all three objectives of cover selection outlined in Section 2 are concurrently achieved. We investigate this phenomenon in Section 3.3.

### 3.1 DDIM-based cover selection

As depicted in Fig. 2, our DDIM approach consists of two steps. We get inspired from DDIM inversion, which refers to the process of using DDIM to achieve the conversion from an image to a latent noise and back to the original image (Kim et al. [2022]).

**Step 1: latent computation.** The initial cover image $\mathbf{x}_0$ (where the subscript denotes the diffusion step) goes through the forward diffusion process described in Eq. 3 to get the latent $\mathbf{x}_T$.

$$\mathbf{x}_{t+1} = \sqrt{\bar{\alpha}_{t+1}} \boldsymbol{f}_{\boldsymbol{\theta}}(\mathbf{x}_t, t) + \sqrt{1 - \bar{\alpha}_{t+1}} \boldsymbol{\epsilon}_{\boldsymbol{\theta}}(\mathbf{x}_t, t) \tag{3}$$

**Step 2: guided image reconstruction**. We optimize $\mathbf{x}_T$ to minimize the loss $||\mathbf{m} - \hat{\mathbf{m}}||$. Specifically, $\mathbf{x}_T$ goes through the backward diffusion process described in Eq. 2 generating cover images that minimize the loss. We evaluate the gradients of the loss with respect to $\mathbf{x}_T$ using backpropagation and use standard gradient based optimizers to get the optimal $\mathbf{x}_T^*$ after some optimization steps.

We use a pretrained DDIM (parametrized by $\theta$), and a pretrained LISO, the state-of-the-art steganographic encoder and decoder from Chen et al. [2022], also described in Appendix A. The weights of the DDIM and the steganographic encoder-decoder are fixed throughout $\mathbf{x}_T$'s optimization process.

The idea is based on the approximation of forward and backward differentials in solving ordinary differential equations (Song et al. [2020]). In the case of deterministic DDIM ($\sigma_t = 0$), Eq. 2 can be used to perform the forward and backward process (Kim et al. [2022]) and achieve accurate image reconstruction. Instead of adopting a fully deterministic DDIM, we find that having a deterministic forward process (Eq. 3) with a stochastic backward process (Eq. 2) yields better results for our setup.

### 3.2 GAN-based cover selection

In the GAN-based approach, we start with a latent vector $\mathbf{z}$ randomly initialized from a Gaussian distribution, which serves as input to the generator $G$. The objective is to identify an optimized $\mathbf{z}^*$ such that the cover image $G(\mathbf{z}^*)$ minimizes the loss $||\mathbf{m} - \hat{\mathbf{m}}||$, i.e.:

$$\mathbf{z}^* = \underset{\mathbf{z}}{\arg\min} ||Dec(Enc(G(\mathbf{z}), \mathbf{m}) - \mathbf{m}|| \tag{4}$$

Where $Enc$, $Dec$ and $\mathbf{m}$ are the steganographic encoder, decoder and secret message respectively as described in Section 2. We evaluate the gradients of the loss with respect to $\mathbf{z}$ using backpropagation and use standard gradient based optimizers to get the optimal $\mathbf{z}^*$ that minimizes the loss. All other modules ($Enc, Dec, G$) are differentiable, pretrained and fixed during the optimization. We utilize BigGAN's pretrained conditional generator (Brock et al. [2018]), and a pretrained LISO steganographic encoder-decoder pair (Chen et al. [2022]).

**Note:** To achieve consistency with the DDIM approach described in Section 3.1, instead of starting with a randomly generated latent vector $\mathbf{z}$, we can begin a cover image $\mathbf{x}$ and apply established GAN inversion techniques to map it to its corresponding latent space (Xia et al. [2022]).

### 3.3 Performance comparison: DDIM & GAN

We compare the performance of the approaches from Sections 3.1 and 3.2 in Table 1. We show the results of 10 randomly selected classes from the ImageNet dataset (Russakovsky et al. [2015]). Following Chen et al. [2022], we assess error rate defined as $\frac{||\mathbf{m} - \hat{\mathbf{m}}||_0}{H \times W \times B}$, the structural similarity index (SSIM) and peak signal-to-noise ratio (PSNR) (Wang et al. [2004]) to measure changes between cover and steganographic images. We further evaluate the generated cover image quality using the no-reference BRISQUE metric (Mittal et al. [2012]). Our methods outperform traditional exhaustive search techniques detailed in Section 1, which are omitted from the table for brevity.

**Methods:** For the DDIM-based cover selection, we generate a batch of 500 cover images, denoted as $\{\mathbf{x}_0^{(i)}\}_{i=1}^{500}$, and apply the cover selection framework to each image independently (Section 3.1). Similarly, for the GAN-based cover selection, we produce a batch of 500 randomly initialized latent vectors, represented as $\{\mathbf{z}^{(i)}\}_{i=1}^{500}$, and independently run the cover selection framework for each vector (Section 3.2). We train a steganographic encoder-decoder pair using 1000 training images from each class, adhering to the method used in Chen et al. [2022]. We then use this trained model, in addition to a diffusion model and BigGAN's conditional generator, both pretrained on ImageNet. We consider a payload of $B = 4$ bpp (we explore different payloads in Section 5.1). The secret messages are random binary bit strings, sampled from an independent $Bernoulli(0.5)$ distribution. We use the binary cross-entropy loss to optimize message recovery. For a comprehensive explanation on our hyperparameter selection, please refer to Appendix B.

**Observation 1:** As shown in Table 1 the optimized images produced by both DDIM and GAN exhibit significantly lower error rates compared to the original images by over **50**% for some classes.

Table 1: Comparative performance of GAN-based and DDIM-based cover selection techniques on the ImageNet dataset, with a payload $B = 4$ bpp. DDIM-optimized images achieve a significant gain over the original images and GAN-optimized images in both error rate reduction and image quality.

| | Error Rate (%) ↓ | | | BRISQUE↓ | | | SSIM↑ | | | PSNR↑ | | |
|---|---|---|---|---|---|---|---|---|---|---|---|---|
| Classes | Original | GAN | DDIM | Original | GAN | DDIM | Original | GAN | DDIM | Original | GAN | DDIM |
| Robin | 2.48 | 1.32 | **1.01** | 27.8 | **18.95** | 19.81 | **0.72** | 0.68 | 0.64 | 22.34 | 23.38 | **23.85** |
| Snow Leopard | 0.84 | **0.36** | 0.55 | 23.71 | 18.28 | **17.26** | **0.75** | 0.74 | 0.72 | 23.71 | 24.54 | **24.96** |
| Daisy | 1.75 | **0.97** | 1.43 | 9.85 | 9.79 | **7.71** | 0.61 | 0.59 | **0.61** | 26.01 | **26.7** | 26.63 |
| Drilling Platform | 2.29 | 1.88 | **1.85** | **25.08** | 25.85 | 27.42 | **0.41** | 0.39 | 0.37 | 21.33 | **21.56** | 21.41 |
| Hartebeest | 0.21 | 0.15 | **0.12** | 16.97 | 16.17 | **13.63** | 0.55 | 0.55 | **0.56** | 24.83 | 25.27 | **26.34** |
| American Egret | 0.95 | **0.77** | 0.78 | 24.4 | 22.9 | **12.03** | 0.63 | 0.63 | **0.64** | 22.72 | 22.87 | **24.49** |
| Owl | 0.21 | **0.02** | 0.09 | 26.01 | 27.77 | **21.3** | 0.73 | **0.76** | 0.71 | 24.02 | 24.62 | **26.01** |
| Chihuahua | 0.79 | 0.59 | **0.55** | 18.45 | 17.92 | **14.33** | 0.58 | 0.56 | **0.59** | 23.13 | 23.55 | **24.44** |
| Cheetah | 2.15 | 2.02 | **1.53** | 41.2 | 40.53 | **35.01** | **0.56** | 0.53 | 0.43 | 21.46 | 21.61 | **21.75** |
| Lady's Slipper | 0.17 | 0.08 | **0.07** | 22.53 | 11.13 | **10.24** | 0.71 | 0.68 | **0.76** | 24.65 | 26.13 | **26.15** |

Surprisingly, although our training objective for cover selection focused solely on minimizing the error rate, we observed improved image quality as evidenced by BRISQUE, SSIM, and PSNR scores. This intriguing relationship between higher image quality and lower error rates is further explored in Appendix H. In summary, our analysis reveals that certain image complexity metrics, including edge density and entropy, negatively correlate with both error rates and BRISQUE scores. This suggests that our cover selection framework modifies features such as edges and entropy during optimization, resulting in enhancements to both image quality and error reduction.

**Observation 2:** DDIM-based optimization consistently outperforms GAN-based methods across all metrics, aligning with previous findings on DDIM's superior image generation capabilities (Dhariwal and Nichol [2021]). We further explore and compare the outputs of both methods, presenting sample steganographic images before and after optimization in Appendix G. Notably, DDIM maintains the semantic integrity of images, preserving key elements like object positions and orientations—such as a bird's unchanged gaze. In contrast, GANs may significantly modify an image's composition, even altering a bird's gaze from left to right, which impacts its semantic content.

***For the remainder of the paper, we will utilize the DDIM-based approach, due to its enhanced performance in both error reduction and image quality.***

## 4   Analysis

In this section, we explore the reasons behind the enhanced performance achieved by our framework. Initially, we analyze the behavior of the pretrained steganographic encoder (Section 4.1). Our observations indicate that the encoder preferentially embeds messages within pixels of low variance. To validate these findings, we compare the encoder's behavior with the waterfilling technique applied to parallel Gaussian channels (Section 4.2). Lastly, we demonstrate that the cover selection optimization effectively increases the presence of low variance pixels. This adjustment equips the encoder with greater flexibility to hide messages, thereby improving overall performance (Section 4.3). We present the results for the ImageNet Robin class with a payload of $B = 4$ bpp. Additional results for various classes and datasets are presented in Appendix D.

### 4.1   Encoding in low-variance pixels

We begin by investigating the underlying mechanism of the pretrained steganographic encoder (Chen et al. [2022]). We hypothesize that the encoder preferentially hides messages in regions of low pixel variance. To test this hypothesis, we structure our analysis into two steps.

**Step 1: variance analysis.** In Fig. 3 (top), we illustrate the variance of each pixel position for the three color channels, calculated across a batch of images and normalized to a range between 0 and 1, as detailed in Appendix D. The plot reveals significant disparities in variance, with certain regions displaying notably lower variance compared to others.

**Step 2: residual computation.** Using the same batch of images, we pass them through the steganographic encoder to obtain the corresponding steganographic images. We then compute the residuals by calculating the absolute difference between the cover and steganographic images and averaging

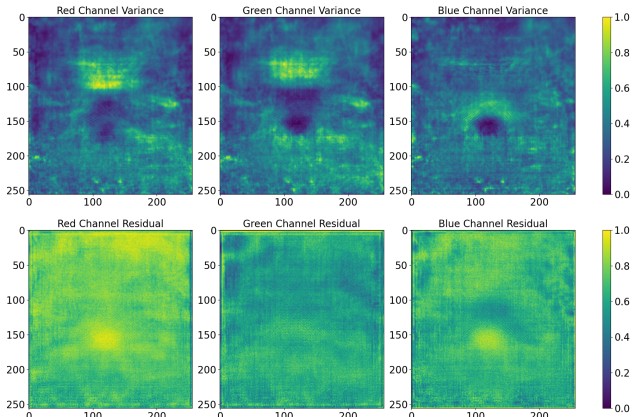

Figure 3: Normalized pixel variances (top) and residuals (bottom) calculated across a batch of 500 Robin images for each color channel, before optimization.

these differences across the batch. This process yields three maps, one for each color channel, which are subsequently normalized to a range between 0 and 1. Those maps are plotted in Fig. 3 (bottom).

As shown in Fig. 3, we observe correlations between the variance and the magnitude of the residual values; where pixels with lower-variance tends to have higher residual magnitudes. To quantify this observation, we introduced a threshold value of 0.5. In the residual maps (from Step 2), locations exceeding this threshold are classified as "high-message regions" and assigned a value of 1. Conversely, locations in the variance maps (from Step 1) falling below this threshold are defined as "low-variance regions", also set to 1. We discovered that **81.6%** of the high-message regions coincide with low-variance pixels. This substantial overlap confirms our hypothesis and underscores the encoder's tactic of utilizing low-variance areas to embed messages, which is a highly desired and natural behavior. We highlight that we are the first to make this observation, despite there being several relevant works on learning-driven steganography; none of these prior studies conducted an interpretation analysis of the encoder to uncover this behavior.

Interestingly, we find that the learned message embedding behavior closely aligns with the water-filling strategy, the theoretically optimal embedding strategy for parallel Additive Gaussian Noise channels, a fundamental concept in communication theory (Cover [1999]). This strategy involves embedding more messages in lower-variance pixel positions, which increases message recovery accuracy. Surprisingly, steganography methods tend to adopt this strategy implicitly, without explicit training to do so. In the subsequent section, we delve deeper into this analogy and further demonstrate the relationship between these two processes.

## 4.2    Analogy to waterfilling

To validate the findings presented in Section 4.1, we draw parallels between our analysis and the waterfilling problem for Gaussian channels. We conceptualize the process of hiding secret messages as transmitting information through $N$ parallel communication channels, where $N$ corresponds to the number of pixels in an image. In this analogy, each pixel operates as an individual communication link, with the secret message functioning as the signal to be hidden and later recovered. The cover image, which embeds the hidden message, serves as noise unknown to the decoder.

We consider a simple additive steganography scheme: $s_i = x_i + \gamma_i m_i$, for $i = 1, 2, ..., N$, where $N = H \times W \times 3$ is the image dimension, $m_i = \{-1, 1\}$ indicates the $i$-th message to be embedded, $\gamma_i$ its corresponding power, $x_i$ and $s_i$ represent the $i$-th element of the cover and steganographic images respectively. We assume a power constraint $P$ that restricts the deviation between the cover and steganographic images: $E\left[\sum_{i=1}^{N}(s_i - x_i)^2\right] \leq P$.

This formulation is similar to the waterfilling solution for $N$ parallel Gaussian channels (Cover [1999]), where the objective is to distribute the total power $P$ among the $N$ channels so as to maximize the capacity $C$, which is maximum rate at which information can be reliably transmitted

over a channel, defined as: $C = \sum_{i=1}^{N} \log_2 \left(1 + \frac{\gamma_i^2}{\sigma_i^2}\right)$, where $\sigma_i^2$ is the variance of $x_i$. The problem can be formulated as a constrained optimization problem, where the optimal power allocation is given by $\gamma_i^2 = \left(\frac{1}{\lambda \ln(2)} - \sigma_i^2\right)^+$, where $(x)^+ = \max(x, 0)$ and $\lambda$ is chosen to satisfy the power constraint.

We calculate $\{\sigma_i^2\}_{i=1}^{3 \times H \times W}$ using a batch of images, and find the optimized $\{\gamma_i^2\}_{i=1}^{3 \times H \times W}$ using the approach described above. We plot the $\gamma_i$'s for each color channel in Fig. 4.

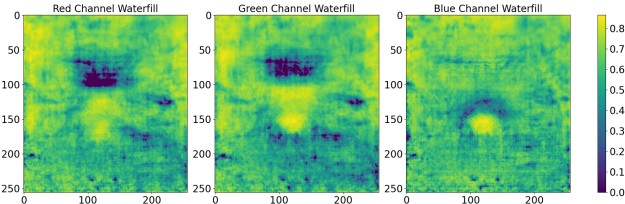

Figure 4: Power coefficients $\gamma_i$ for each color channel, calculated using a batch of 500 Robin images.

We observe a degree of similarity when comparing with Fig. 3 (bottom). To quantitatively assess this resemblance across color channels, we quantize the three matrices by setting values greater than 0.5 to 1 and values less than 0.5 to 0. For each channel, the similarity is calculated using the equation $\frac{\sum_{i,j} \mathbf{1}(\mathbf{W}_{ij}^{(k)} = \mathbf{R}_{ij}^{(k)})}{256 \times 256}$, where $\mathbf{W}_{ij}^{(k)}$ and $\mathbf{R}_{ij}^{(k)}$ are the $(i,j)$-th pixels of the quantized waterfilling and residual matrices, respectively, for the channel $k$. The computed similarity scores are **81.8%** for red, **65.5%** for green, and **74.9%** for blue, revealing varying degrees of resemblance with the waterfilling strategy across the color channels. The variation underscores that the waterfilling strategy is implemented more effectively in some channels than in others.

## 4.3 Impact of cover selection

A natural question becomes: what is the cover selection optimization doing? We plot the variance maps of the optimized cover images in Fig. 5.

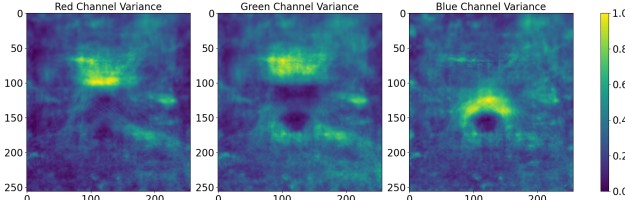

Figure 5: Normalized pixel variances across a batch of 500 Robin images for each color channel, after optimization.

We notice that the number of low variance spots significantly increased as compared to Fig. 3 (top), meaning that the encoder has more freedom in encoding the secret message. Quantitatively, we find that **92.4%** of the identified high-message positions are encoded in low-variance pixels, as compared to **81.6%** before optimization. Given that the encoder preferentially embeds data in these low variance areas, this increase provides greater flexibility for data embedding, thereby explaining the performance gains observed in our framework.

## 5 Practical settings

In this section, we adapt our framework for practical considerations. We evaluate its performance across different payloads (Section 5.1), adapt it for JPEG compression (Section 5.2), and confirm security against steganalysis (Section 5.3). Computational times are detailed in Appendix I. We use two datasets, CelebA-HQ (Karras et al. [2017]) and AFHQ-Dog (Choi et al. [2020]), using the same settings described in Section 3.3.

Table 2: Performance comparison across AFHQ-Dog and CelebA-HQ across various payloads. We observe that the error rates of DDIM-optimized images are significantly lower than original images.

| | Payload $B$ | Error Rate (%) ↓ | | BRISQUE ↓ | | SSIM ↑ | | PSNR ↑ | |
| | | Original | DDIM | Original | DDIM | Original | DDIM | Original | DDIM |
|---|---|---|---|---|---|---|---|---|---|
| CelebA-HQ | 1 bpp | 2.6E-04 | **1.5E-05** | **2.75** | 4.07 | **0.95** | 0.94 | 36.25 | **36.37** |
| | 2 bpp | 2.3E-03 | **9E-04** | **5.9** | 9.7 | 0.91 | **0.92** | 31.82 | **32.46** |
| | 3 bpp | 0.011 | **0.002** | 9.95 | **9.83** | 0.86 | **0.87** | 32.16 | **33.88** |
| | 4 bpp | 0.051 | **0.019** | 11.91 | **11.04** | 0.81 | **0.83** | 30.91 | **32.46** |
| AFHQ-Dog | 1 bpp | 8E-05 | **0.00** | 12.14 | **12.11** | **0.94** | 0.93 | 36.84 | **36.87** |
| | 2 bpp | 8E-04 | **6.8E-05** | **4.12** | 7.19 | 0.93 | **0.94** | **35.1** | 34.4 |
| | 3 bpp | 0.007 | **0.002** | 10.34 | **6.87** | **0.86** | 0.85 | 32.5 | **32.6** |
| | 4 bpp | 0.11 | **0.09** | 13.49 | **13.42** | 0.75 | **0.76** | 28.97 | **28.99** |

## 5.1 Payload impact on performance

We explore different payload capacities $B$, highlighted in Table 2. We show the results for $B = 1, 2, 3, 4$ bits per pixel (bpp). DDIM-optimized images show error rates significantly lower than originals, with image quality metrics like BRISQUE, SSIM, and PSNR largely preserved, though some quality decline was noted at lower bpp levels in CelebA-HQ and AFHQ-Dog. We include sample generared cover images generated using the DDIM framework in Appendix E. Despite experimenting with various regularization techniques aimed at maintaining image quality, no noticeable improvement was observed (Appendix C). Considering this, extending our framework to explore novel regularization techniques for such payload capacities is an interesting future direction. We also provide example cover and steganographic images generated by the LISO framework under different payload values in Appendix F.

## 5.2 JPEG compression

Robustness against lossy image compression is crucial for steganography. We extend our framework to accommodate JPEG compression (Wallace [1991]). Following Athalye et al. [2018], we implement an approximate JPEG layer where the forward pass executes standard JPEG compression, while the backward pass operates as an identity function. Once the encoder-decoder pair is trained, we generate a JPEG-compliant cover image following the framework described in Section 3.1, augmented by adding a JPEG layer post-encoding. In Table 3, we demonstrate that our framework achieves improved error rates for $B = 1$ bpp, thereby validating our approach's capability to optimize cover images under JPEG compression constraints. In addition, we show robustness results to Gaussian noise in Appendix K.

Table 3: JPEG results for $B = 1$ bpp.

| | Error Rate % ↓ | | PSNR ↑ | |
| Dataset | Original | DDIM | Original | DDIM |
|---|---|---|---|---|
| CelebA-HQ | 0.12 | **0.06** | 21.09 | **21.53** |
| AFHQ-Dog | 0.15 | **0.11** | 19.34 | **19.63** |

Table 4: Steganalysis results AFHQ-Dog.

| | Payload $B$ | Error Rate (%) ↓ | | XuNet Det. (%) ↓ | |
| | | Original | DDIM | Original | DDIM |
|---|---|---|---|---|---|
| Scenario 1 | 1 bpp | 8E-05 | **0.00** | **37.1** | 37.5 |
| | 2 bpp | 8E-04 | **6.8E-05** | 31.34 | **15.42** |
| | 3 bpp | 0.007 | **0.002** | **20.39** | 34.82 |
| | 4 bpp | 0.11 | **0.09** | 97.37 | **97.35** |
| Scenario 2 | 1 bpp | 0.0026 | **2E-05** | 0.0 | 0.0 |
| | 2 bpp | 0.0024 | **1E-04** | 0.0 | 0.0 |
| | 3 bpp | 0.01 | **0.003** | 3.2 | **2.1** |
| | 4 bpp | 0.23 | **0.22** | 9.2 | **8.6** |

## 5.3 Steganalysis

Steganalysis systems are designed to detect whether there is hidden information within images. As these tools evolve, neural steganography techniques now integrate these systems into their end-to-end pipelines to create images that can bypass detection (Chen et al. [2022], Shang et al. [2020]). We show our results in Table 4 on the AFHQ-Dog dataset. Following the approach in Chen et al. [2022], we evaluate the security of our optimized images by measuring the detection rate using the steganalysis tool XuNet (Xu et al. [2016]) and also record message recovery error rates. The image quality metrics, such as BRISQUE, SSIM, and PSNR, are comparable to those listed in Table 2 and have therefore been omitted for brevity. We explore two different scenarios:

**Scenario 1:** In this scenario, the experimental setup remains the same as described in Section 3.1 and illustrated in Fig. 2. The steganographic encoder-decoder pair is trained without regularizers to evade steganalysis detection. The DDIM-optimized images exhibit comparable detection rates at payloads of $B = 1$ and $B = 4$, superior performance at $B = 2$, and inferior performance at $B = 3$, all while achieving significantly lower error rates. While it is puzzling that detection rates do not consistently decrease with lower payload size, this phenomenon is also observed in LISO Chen et al. [2022], on which our framework is built. We provide a more detailed discussion in Appendix J.

**Scenario 2:** We leverage the differentiability of XuNet as described in Chen et al. [2022]. During the optimization of the steganographic encoder-decoder pair, we introduce an additional loss term to account for steganalysis. This adjustment leads to a notable reduction in detection rates across all payload sizes, while maintaining consistently low error rates for both original and DDIM-optimized images. Notably, DDIM-optimized images exhibit even lower detection and error rates compared to the original images, demonstrating superior performance.

Further implementation details, along with results using an alternative steganalysis method, SRNet (Boroumand et al. [2018]), are provided in Appendix J.

# 6    Conclusion

We propose a novel cover selection framework for steganography leveraging pretrained generative models. We demonstrate that by carefully optimizing the latent space of these models, we generate steganographic images that exhibit high visual quality and embedding capacity. Additionally, our information-theoretic analysis shows that message hiding predominantly occurs in low-variance pixels, reflecting the waterfilling algorithm's approach to parallel Gaussian channels. Our framework is versatile, allowing for the incorporation of further constraints to produce JPEG-resistant steganographic images or to evade detection by particular steganalysis systems. For future work, we aim to expand our analysis (Section 4.2) to draw similarities with correlated Gaussian channels, moving beyond the independent channels considered in this work.

# Acknowledgments

This work was partly supported by ARO Award W911NF2310062, ONR Award N000142412542, and the 6G@UT center within the Wireless Networking and Communications Group (WNCG) at the University of Texas at Austin.

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

# A   Learned Iterative Steganography Optimization (LISO)

LISO (Chen et al. [2022]) advances the method established in Kishore et al. [2021], which is centered around Optimization-based Image Steganography. Leveraging a differentiable decoder equipped with either randomly initialized or pretrained weights (as referenced in the preceding paragraph), Kishore et al. [2021] formulates the steganography encoding as an optimization task for each sample. This approach is similar to the generation of adversarial perturbations as discussed in Szegedy et al. [2013]. Specifically, the technique described in Kishore et al. [2021] seeks to compute a steganographic image by addressing a constrained optimization problem that ensures the perturbed image remains within the bounds of the $[0, 1]^{H \times W \times 3}$ hypercube.

$$\min_{\mathbf{s} \in [0,1]^{H \times W \times 3}} L_{\mathrm{acc}}(\mathrm{Dec}(\mathbf{s}), \mathbf{m}) + \lambda L_{\mathrm{qua}}(\mathbf{s}, \mathbf{x}) \tag{5}$$

where

$$L_{\mathrm{acc}}(\hat{\mathbf{m}}, \mathbf{m}) := \langle \mathbf{m}, \log \hat{\mathbf{m}} \rangle + \langle (1 - \mathbf{m}), \log(1 - \hat{\mathbf{m}}) \rangle \tag{6}$$

$$L_{\mathrm{qua}}(\mathbf{s}, \mathbf{x}) := \frac{1}{N} \|\mathbf{s} - \mathbf{x}\|^2, \tag{7}$$

where $\mathbf{m}$ represents the secret message, $\hat{\mathbf{m}}$ is the decoded message, $\mathbf{x}$ is the cover image, and $\mathbf{s}$ is the steganographic image. The operation $\langle \cdot \rangle$ signifies the dot product, $\lambda$ is a scaling factor, and $N = H \times W \times 3$ represents the total number of pixels in the image, with $H$ and $W$ being the height and width of the image, respectively. The accuracy loss, $L_{\mathrm{acc}}(\hat{\mathbf{m}}, \mathbf{m})$, is calculated using binary cross entropy to minimize the distance between the estimated and actual messages, while the quality loss, $L_{\mathrm{qua}}(\mathbf{s}, \mathbf{x})$, employs mean squared error to ensure the steganographic image closely resembles the cover image. This objective function is represented as $\ell(\mathbf{x}, \mathbf{m})$. To solve the optimization problem outlined above, various solvers can be utilized and as shown in Algorithm 1, with iterative, gradient-based algorithms. In Algorithm 1, $\eta > 0$ is the step size, and $g(\cdot)$ describes the update function specific to the optimization method used. The perturbation $\delta$ is iteratively adjusted to minimize the loss $\ell$ while adhering to the pixel constraints of the image.

---

**Algorithm 1** Iterative Optimization

---

1: $\delta_0 \leftarrow 0$
2: **for** $t = 1$ **to** $T$ **do**
3: $\quad \delta_t \leftarrow \delta_{t-1} + \eta \cdot g\left(\nabla_\delta \ell(\mathbf{x} + \delta_{t-1}, \mathbf{m}), \mathbf{x}, \delta_{t-1}\right)$
4: **end for**
5: $\mathbf{s} \leftarrow \mathbf{x} + \delta_T$

---

In LISO, The function $g(\cdot)$ in Algorithm 1 is approximated using a fully convolutional network designed around a gated recurrent unit (GRU). The complete LISO framework, which includes the iterative encoder, decoder, and critic, undergoes end-to-end training on a diverse image dataset. Similar to the training process of Generative Adversarial Networks (GANs), the training of LISO alternates between optimizing the critic and the encoder-decoder networks. Throughout this training phase, losses for all intermediate updates are calculated with exponentially increasing weights $\left(\gamma^{T-t}\right.$ at step $t$). With intermediate predictions denoted as $\hat{\mathbf{m}}_1, \ldots, \hat{\mathbf{m}}_T$ the loss is:

$$L_{\mathrm{train}} = \sum_{t=1}^{T} \gamma^{T-t} \left[ L_{\mathrm{acc}}(\mathbf{m}, \hat{\mathbf{m}}_t) + \lambda L_{\mathrm{qua}}(\mathbf{x}, \mathbf{s}_t) + \mu L_{\mathrm{crit}}(\mathbf{x}, \mathbf{s}_t) \right],$$

where $\gamma \in (0, 1)$ is a decay factor and $L_{\mathrm{crit}}$ denotes the critic loss to generate real-looking images (with weight $\mu > 0$).

# B   Training details

## B.1   GAN-based cover selection

In our GAN-based cover selection method, we utilize the BigGAN generator (Brock et al. [2018]) and a LISO encoder-decoder pair (Chen et al. [2022]), both pretrained on the ImageNet dataset

(Russakovsky et al. [2015]). Specifically, the BigGAN generator receives a latent vector $\mathbf{z}$, a 128-dimensional vector initialized from a truncated normal distribution with truncation set at 0.4, and a class index $c$. It then produces the cover image $\mathbf{x} \in [0,1]^{H \times W \times 3}$. The LISO encoder processes $\mathbf{x}$ along with the secret message $\mathbf{m} \in \{0,1\}^{H \times W \times B}$ to create the steganographic image $\mathbf{s}$, while the LISO decoder attempts to recover $\hat{\mathbf{m}}$ from $\mathbf{s}$. We consider a payload $B = 4$. To optimize the latent vector $\mathbf{z}$, we minimize the binary cross-entropy loss $BCE(\mathbf{m}, \hat{\mathbf{m}})$ using the Adam optimizer with a learning rate of 0.01 over 100 epochs. Both the GAN generator and the LISO encoder-decoder are configured with the same architecture and parameters as described in their respective original publications.

To replicate the results presented in Table 1, we optimize a batch of 500 latent vectors $\{\mathbf{z}^{(i)}\}_{i=1}^{500}$ for each class. These vectors are randomly initialized and subsequently optimized. We then report several metrics: the average error rate between the original message $\mathbf{m}$ and the estimated message $\hat{\mathbf{m}}$, the average BRISQUE scores of the cover images to assess their naturalness, and both the SSIM (Structural Similarity Index) and PSNR (Peak Signal-to-Noise Ratio) values to evaluate the similarity and quality between the cover and steganographic image pairs.

### B.2   DDIM-based cover selection

In our cover selection method based on Denoising Diffusion Implicit Models (DDIM), we employ three DDIM models alongside LISO encoder-decoder pairs, each pretrained on different datasets: ImageNet (Russakovsky et al. [2015]), AFHQ-Dog (Choi et al. [2020]), and CelebA-HQ (Karras et al. [2017]).

Following the procedure outlined in Section 3.1, we initiate the process by sampling a random image $\mathbf{x_0}$. We then execute the deterministic forward DDIM process over $T$ steps, with each step defined as follows:

$$\mathbf{x}_{t+1} = \sqrt{\bar{\alpha}_{t+1}}\boldsymbol{f_\theta}(\mathbf{x}_t, t) + \sqrt{1 - \bar{\alpha}_{t+1}}\boldsymbol{\epsilon_\theta}(\mathbf{x}_t, t) \tag{8}$$

After obtaining the latent representation $\mathbf{x}_T$, we initiate the stochastic reverse DDIM process, which spans $E$ epochs. Within each epoch, we perform the reverse DDIM process on the acquired latent for $N$ iterations. Each iteration proceeds as follows:

$$\mathbf{x}_{t-1} = \sqrt{\bar{\alpha}_{t-1}}\boldsymbol{f_\theta}(\mathbf{x}_t, t) + \sqrt{1 - \bar{\alpha}_{t-1} - \sigma_t^2}\boldsymbol{\epsilon_\theta}(\mathbf{x}_t, t) + \sigma_t^2\boldsymbol{\epsilon} \tag{9}$$

Where $\boldsymbol{\epsilon_\theta}$ is a pretrained network, $\boldsymbol{f_\theta}$ is a function of $\boldsymbol{\epsilon_\theta}$, $\sigma_t^2 = \sqrt{0.5 \cdot \left(\left(1 - \frac{\bar{\alpha}_t}{\bar{\alpha}_{t-1}}\right) \cdot \frac{1 - \bar{\alpha}_{t-1}}{1 - \bar{\alpha}_t}\right)}$, and $\boldsymbol{\epsilon} \sim \mathcal{N}(\mathbf{0}, \mathbf{I})$.

We configure our model with the following parameters: $E = 50$ epochs, $T = 40$ time steps, and $N = 6$ iterations per epoch. For optimization, we employ the Adam optimizer with a learning rate of $2E - 06$. The variance schedule that determines $\bar{\alpha}_t$ and $\bar{\alpha}_{t-1}$, as well as the DDIM architectures, are consistent with those described in Kim et al. [2022].

## C   Regularization effect

Despite testing several regularization methods intended to preserve image quality—including total variation (Rudin et al. [1992]), edge preservation (Perona and Malik [1990]), feature matching with a pre-trained VGG network (Gatys et al. [2015]), and a classic $l_1$ distance between the original and updated cover images—we observed no significant enhancements. These results are shown in Table 5.

## D   Encoding operation analysis: additional results

In this section, we further describe the encoder's strategy of embedding messages in regions with low pixel variance, as described in Section 4.1.

Table 5: Performance results with regularization on CelebA-HQ with a payload $B = 2$ bpp. We show the resuls for edge preservation (EP), $l_1$ distance between original and updated cover images ($l_1$), total variation (TV), and VGG feature matching (VGG).

| | Error Rate (%) ↓ | | BRISQUE↓ | | SSIM↑ | | PSNR↑ | |
|---|---|---|---|---|---|---|---|---|
| Method | Original | DDIM | Original | DDIM | Original | DDIM | Original | DDIM |
| EP | 2E-03 | 7E-04 | 11.62 | 13.25 | 25.57 | 25.65 | 0.85 | 0.85 |
| $l_1$ | 2E-03 | 1E-03 | 11.94 | 13.45 | 25.65 | 25.7 | 0.85 | 0.85 |
| TV | 2E-03 | 7E-04 | 11.58 | 13.43 | 25.48 | 25.57 | 0.85 | 0.85 |
| VGG | 2.1E-03 | 7.5E-04 | 11.32 | 13.65 | 25.59 | 25.65 | 0.85 | 0.85 |

We calculate the variance for each pixel position across a batch of 500 images, separately for each of the three color channels. This results in three variance maps, each of shape 256x256 (corresponding to the dimensions of the images). These variance maps are normalized to a range between 0 and 1 to facilitate subsequent analysis.

We present variance and residual maps for three additional ImageNet classes: Daisy (Fig. 6), Yellow Lady's Slipper (Fig. 7), and American Egret (Fig. 8). These visualizations validate that the encoder predominantly conceals messages within areas of low variance. Further analysis includes the CelebA-HQ dataset with a payload of $B = 2$ bpp, illustrated in Fig. 9. Notably, in the Blue channel, the encoder distinctly favors low variance pixels for message concealment. Intriguingly, in the Green channel, regions such as the eyes, nose, and mouth are preferred for embedding messages. Investigating the underlying reasons for this selective use is an interesting open problem.

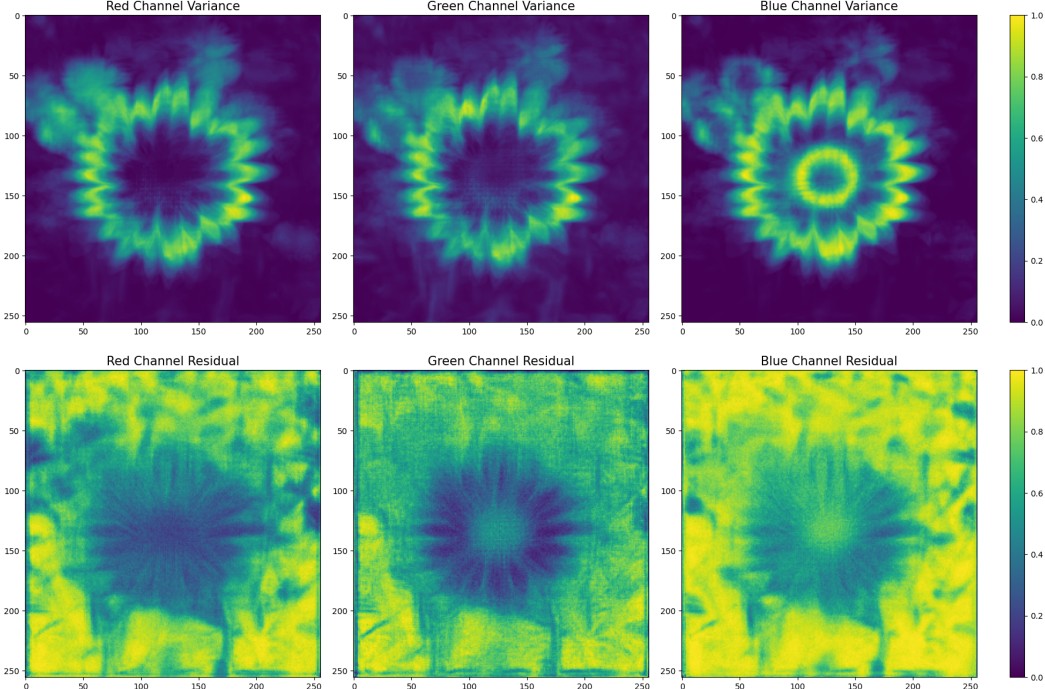

Figure 6: Normalized pixel variances (top) and residuals (bottom) across a batch of 500 images for the ImageNet Daisy class.

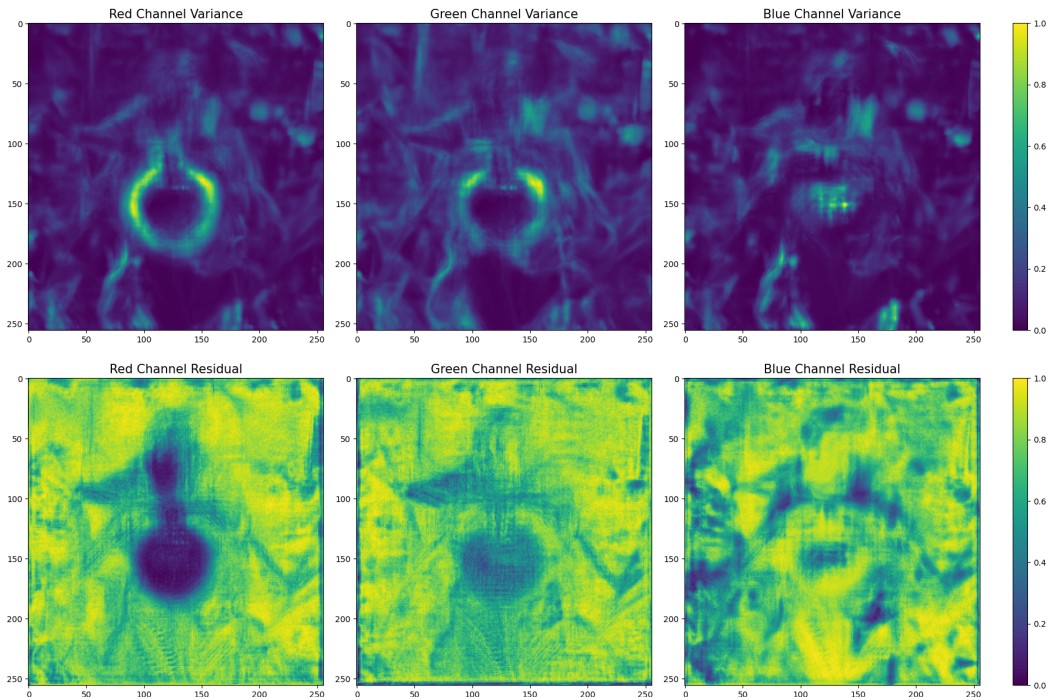

Figure 7: Normalized pixel variances (top) and residuals (bottom) across a batch of 500 images for the ImageNet Yellow Lady's Slipper class.

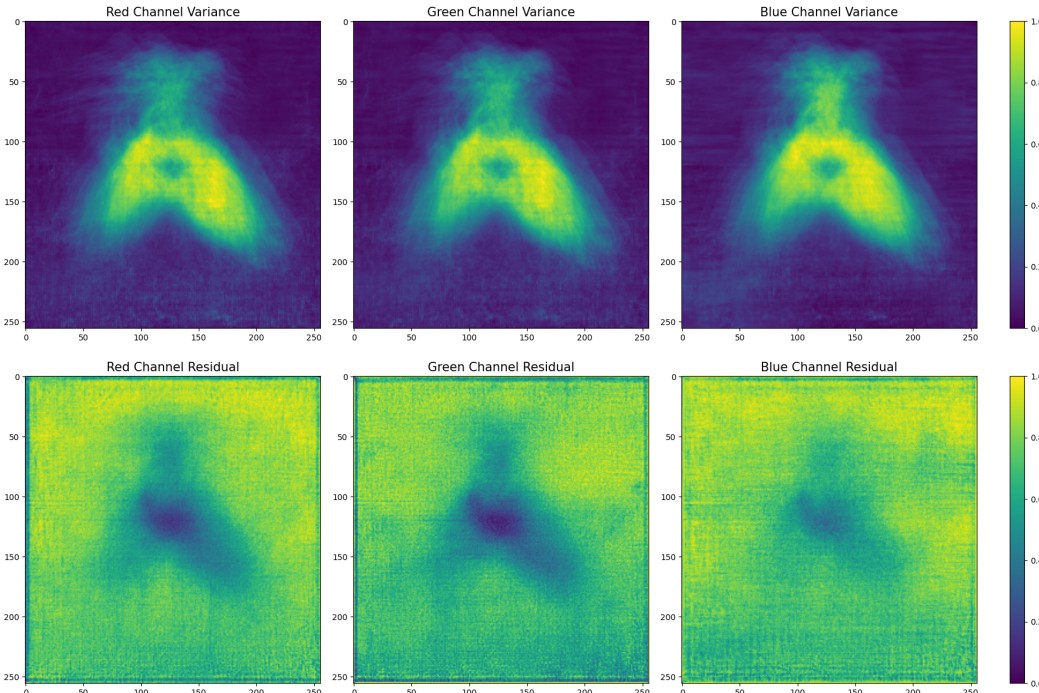

Figure 8: Normalized pixel variances (top) and residuals (bottom) across a batch of 500 images for the ImageNet American Egret class.

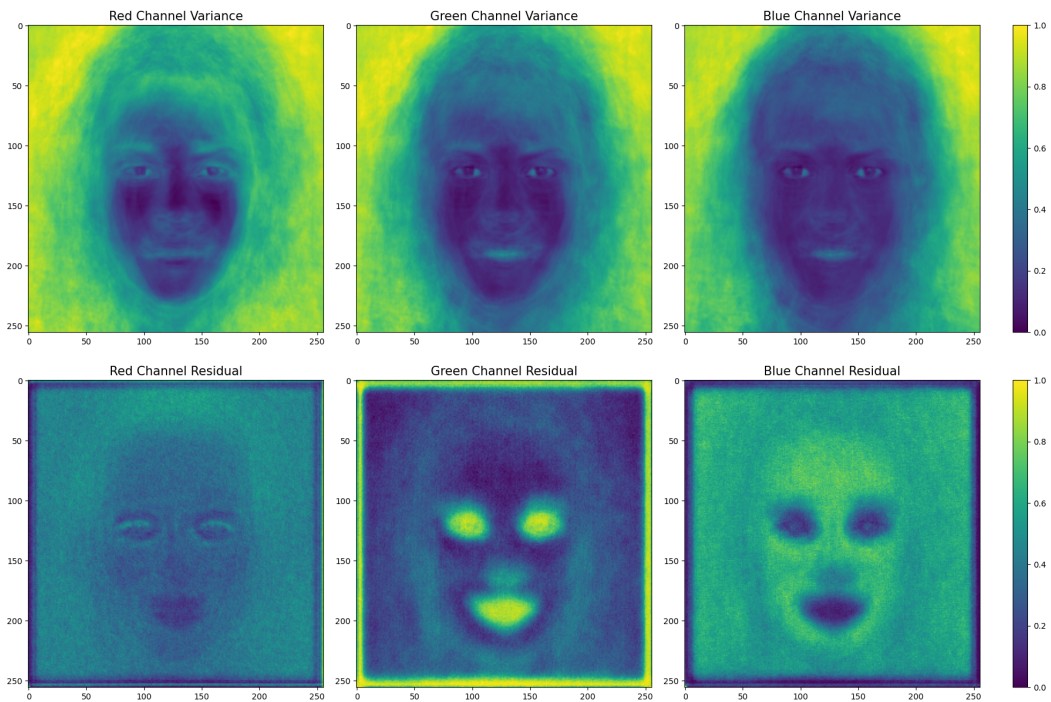

Figure 9: Normalized pixel variances (top) and residuals (bottom) across a batch of 500 images for the CelebA-HQ dataset.

# E    DDIM sample cover images

In this section, we present optimized cover images generated by our DDIM cover selection framework. Samples from both the CelebA-HQ and AFHQ-Dog datasets are displayed, showcasing variations for different payload capacities with $B = 1, 2, 3, 4$ bits per pixel (bpp).

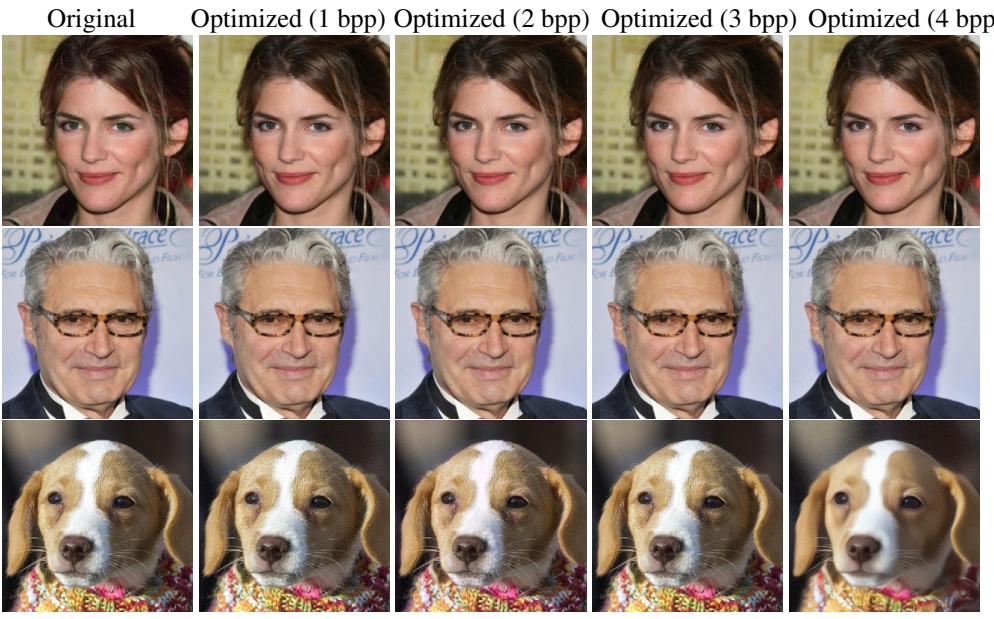

Figure 10: Generated DDIM cover images for different message payload values.

# F    Sample steganographic images

In this section, we present a selection of randomly sampled cover images from CelebA-HQ and AFHQ alongside their steganographic counterparts generated using the LISO framework (Chen et al. [2022]). The results are demonstrated for various payload capacities, ranging from $B = 1$ to $4$ bits per pixel (bpp).

| Cover image | 1 bpp | 2 bpp | 3 bpp | 4 bpp |
| --- | --- | --- | --- | --- |

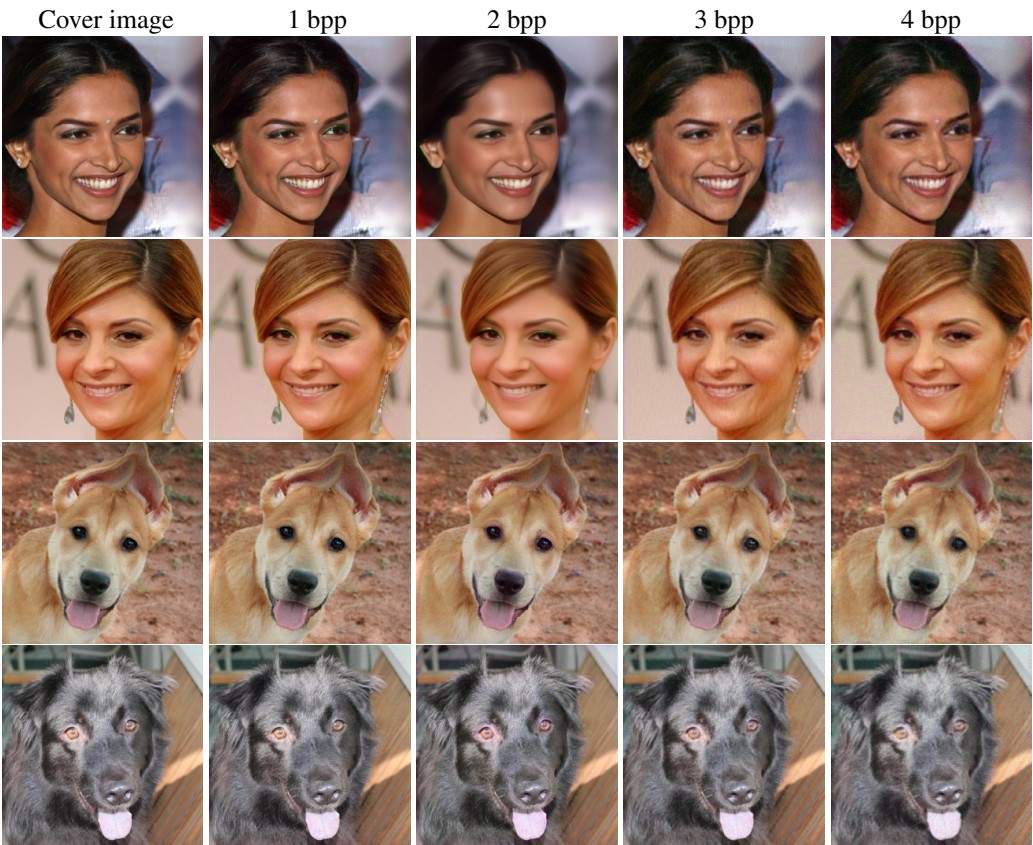

Figure 11: Covers and their corresponding steganographic images.

# G    Sample steganographic images: DDIM vs GAN

We compare the outputs of both methods, presenting sample steganographic images before and after optimization in Fig 12 for a payload $B = 4$ bits per pixel. DDIM conserves the semantic essence of images, maintaining critical aspects such as the positions and orientations of objects—for instance, a bird's gaze remains consistent. In contrast, GANs can substantially alter an image's structure, potentially changing a bird's gaze direction, thus affecting its semantic meaning.

# H    Image complexity metrics

In this section, we explore the intriguing observation that optimizing for error rate not only preserves image quality but, in some instances, even enhances it. This occurs despite the fact that our primary focus is not directly on image quality optimization.

We assess various complexity metrics—entropy, edge density, compression ratio, and color diversity—across a dataset of 500 images from the AFHQ-Dog collection, each embedded with a payload of $B = 4$ bits per pixel. Our analysis investigates how these metrics correlate with the message error rate, as depicted in Figure 13. Furthermore, we investigate the relationship between these complexity metrics and the BRISQUE image quality score, as shown in Figure 14.

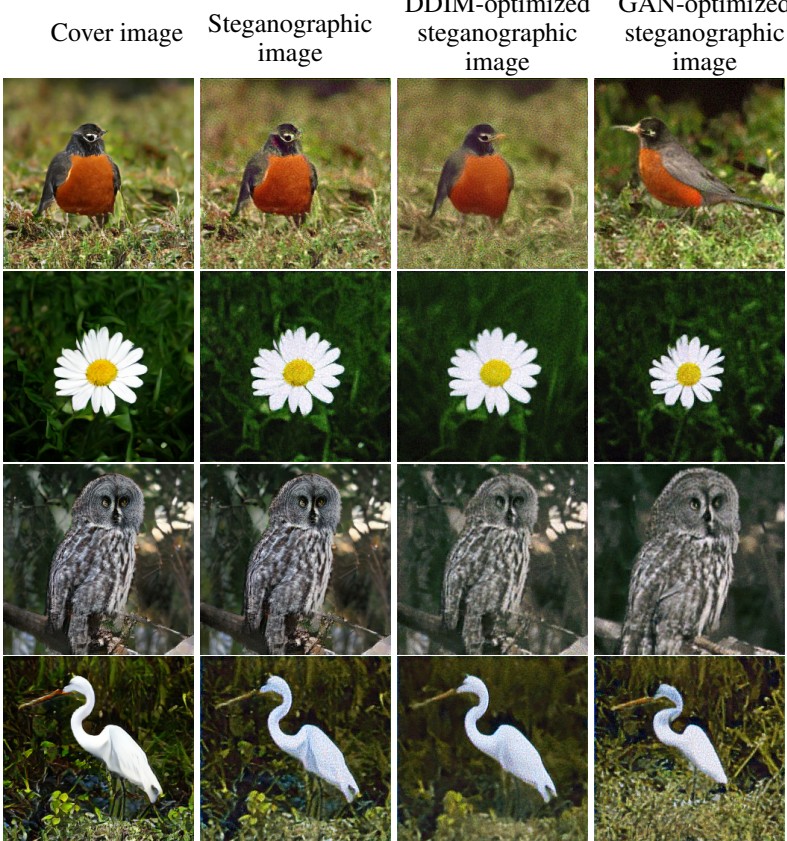

Figure 12: Generated steganographic images: GAN vs DDIM.

**The entropy** of an image measures the randomness of intensity values and is calculated as $-\sum_{i=1}^{256} p_i \log_2 p_i$, where $p_i$ is the probability of occurrence of the $i^{th}$ intensity value, calculated over a batch images. The probabilities are determined from the grayscale version of each image $C$, where the grayscale conversion simplifies the entropy calculation by focusing on the luminance information while discarding color details.

**The edge density** of an image measures the proportion of pixels that are part of edges to the total number of pixels in the image. This is typically calculated by first applying an edge detection algorithm, such as the Sobel or Canny operator, to identify edge pixels. The edge density is then quantified as $\frac{n_e}{N}$, where $n_e$ is the number of edge pixels identified, and $N$ is the total number of pixels in the image.

**The compression ratio** of an image is a measure of the efficiency of a compression algorithm, defined as the ratio of the original file size to the compressed file size. Mathematically, it can be expressed as $\frac{S_{original}}{S_{compressed}}$ where $S_{original}$ is the original file size in bytes, and $S_{compressed}$ is the size of the file after compression. A higher compression ratio indicates more effective compression, reducing storage and transmission resource requirements.

**The color diversity** in an image refers to the variety and distribution of colors present within the image. It can be quantified by analyzing the image's color histogram, which represents the frequency of each color in the image. Color diversity is often measured using metrics such as the number of distinct colors, or the evenness of their distribution. A common approach is to calculate the Shannon diversity index, expressed as $-\sum_{i=1}^{k} p(c_i) \log_2 p(c_i)$, where $p(c_i)$ denotes the proportion of pixels of color $c_i$ and $k$ is the total number of unique colors in the histogram. High color diversity indicates a rich variety of colors, which typically contributes to the visual complexity and aesthetic quality of the image.

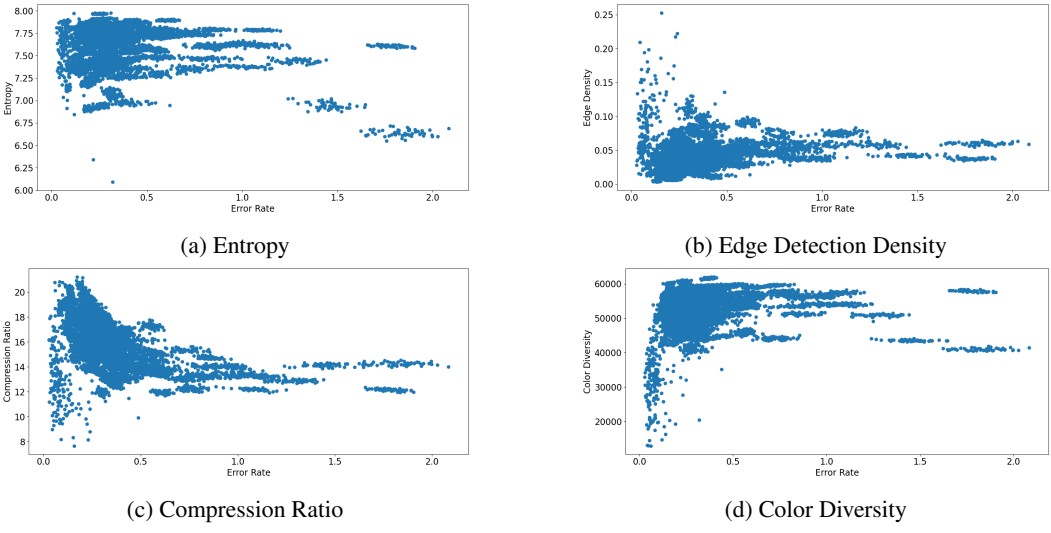

(a) Entropy

(b) Edge Detection Density

(c) Compression Ratio

(d) Color Diversity

Figure 13: Image complexity metrics vs error rate

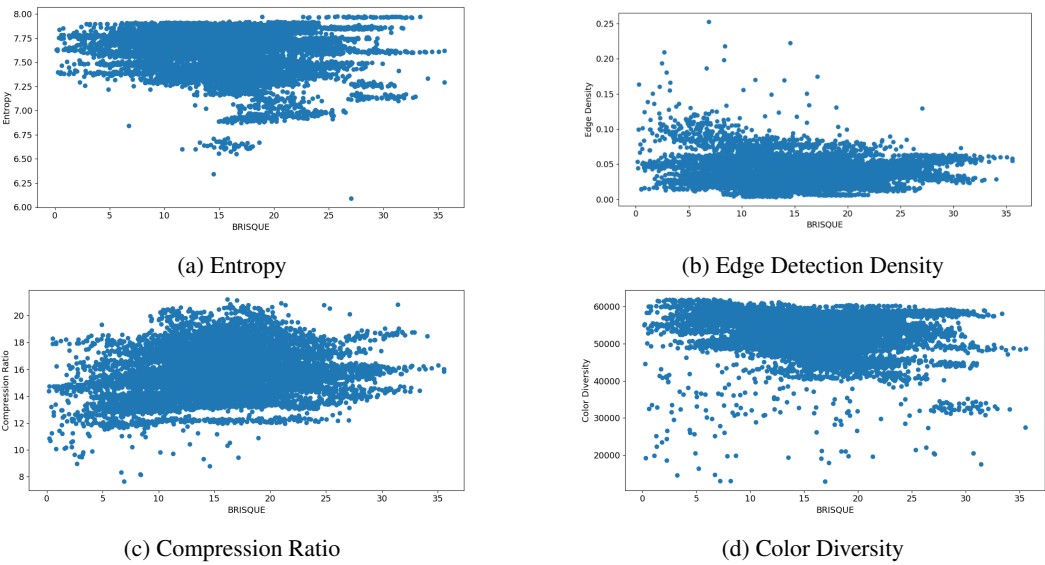

(a) Entropy

(b) Edge Detection Density

(c) Compression Ratio

(d) Color Diversity

Figure 14: Image complexity metrics vs BRISQUE

While we do not observe a perfect correlation across all metrics, we note consistent trends with certain measures such as entropy and edge density. Specifically, as entropy and edge density increase, both the error rate and BRISQUE scores tend to decrease. We hypothesize that optimizing for error rate influences certain image characteristics, such as entropy and edge density, which are also associated with BRISQUE scores. This relationship may partially explain why we observe improved BRISQUE scores even though our primary focus is on minimizing error rate. For the other metrics, namely compression ratio and color diversity, the patterns are not as clear.

# I Computational time

We show the average time required to optimize images using the DDIM-based cover selection method in Table 6 for both the CelebA-HQ and AFHQ-Dog datasets. We calculate computation time by determining the number of DDIM backward sampling steps required to achieve the lowest message recovery error, and then multiplying that number by the average duration of each step. As anticipated, for the CelebA-HQ dataset, the average computation time increases with the payload size. A similar

trend is observed in the AFHQ-Dog dataset; however, an exception occurs at a payload of $B = 4$ bpp. This anomaly can be attributed to the fact that, as shown in Table 2, the DDIM-optimized images for this payload do not exhibit a significantly lower error rate compared to the original images. All experiments were conducted using a NVIDIA A-100 GPU.

Table 6: Average computation time of DDIM-based cover selection (in seconds) for different payload values.

| Dataset | 1 bpp | 2 bpp | 3 bpp | 4 bpp |
|---------|-------|-------|-------|-------|
| CelebA-HQ | 0.69 | 6.85 | 11.52 | 24.83 |
| AFHQ-Dog | 0.04 | 0.91 | 5.33 | 0.34 |

## J  Steganalysis: detailed settings and additional experiments

We adopt the simulation settings outlined in Chen et al. [2022] for our experiments. **In Scenario 1**, the steganography model M is trained without specific techniques to avoid detection by steganalysis. We assume the attacker, who performs steganalysis, knows the architecture of M but has no access to its weights, training data, or hyperparameters. However, the attacker can train a surrogate model M' to generate their own steganographic images. To simulate this scenario, we trained a steganalysis model on the CelebA dataset and used it to detect steganographic images generated from the AFHQ-Dog dataset. Interestingly, detection rates did not consistently decrease with lower payload sizes, a phenomenon also noticed in LISO Chen et al. [2022], on which our framework is based. We hypothesize this behavior arises from the distributional mismatch between training and testing data, as discussed earlier. **In scenario 2**, We leverage the fact that neural steganalysis methods are entirely differentiable, and that LISO uses gradient-based optimization. This allows us to reduce security risk by incorporating an additional loss term from the steganalysis system into the LISO optimization process. Specifically, during evaluation, if an image is identified as steganographic, we add the logit value of the steganographic class to the loss function.

In addition to XuNet (Xu et al. [2016]), we compute the steganalysis results of SRNet, another state-of-the-art steganalysis system (Boroumand et al. [2018]). The results of both schemes are compared in Table 7. Our observations indicate that the images generated by our framework effectively resist steganalysis by SRNet. This is evidenced by the significant drop in detection rate when transitioning from scenario 1 to scenario 2. As a reminder, in scenario 2, we exploit the differentiability of the steganalyzer (SRNet) and incorporate an additional loss term to account for steganalysis.

Table 7: Steganalysis results with SRNet and XuNet.

| | Payload $B$ | SRNet Det. (%) ↓ | | XuNet Det. (%) ↓ | |
|---|-------------|------------------|------|------------------|------|
| | | Original | DDIM | Original | DDIM |
| Scenario 1 | 1 bpp | 15 | **13** | **37.1** | 37.5 |
| | 2 bpp | **14.5** | 32.5 | 31.34 | **15.42** |
| | 3 bpp | 61.5 | **55.5** | **20.39** | 34.82 |
| | 4 bpp | 76 | 76 | 97.37 | **97.35** |
| Scenario 2 | 1 bpp | 0.0 | 0.0 | 0.0 | 0.0 |
| | 2 bpp | 0.0 | 0.0 | 0.0 | 0.0 |
| | 3 bpp | 0.0 | 0.0 | 3.2 | **2.1** |
| | 4 bpp | 2 | **1** | 9.2 | **8.6** |

## K  Robustness to Gaussian noise

In this section, we evaluate the robustness of our DDIM-based approach to Gaussian noise. The experimental setup remains the same as described in Section 3.1 and illustrated in Fig. 2, with the only modification being the injection of Gaussian noise, distributed as $\mathcal{N}(0, \beta)$, into the output of the steganographic encoder. The decoder subsequently processes the noisy steganographic image to estimate the embedded message. Results of this experiment are presented in Table 8. Our findings demonstrate that the proposed framework produces cover images resilient to Gaussian noise,

Table 8: Robustness to Gaussian noise for a payload $B = 4$ bpp on CelebA-HQ.

| Variance $\beta$ | Error Rate (%) ↓ | | BRISQUE ↓ | | SSIM ↑ | | PSNR ↑ | |
|---|---|---|---|---|---|---|---|---|
| | Original | DDIM | Original | DDIM | Original | DDIM | Original | DDIM |
| 0.01 | 2.2 | **1.9** | **12.1** | 12.8 | 0.62 | **0.63** | 26.85 | **27.6** |
| 0.02 | 7.1 | **6.5** | 15.98 | **14.33** | **0.57** | 0.56 | 25.75 | **26.34** |
| 0.03 | 12.3 | **11.8** | 15.01 | **14.38** | 0.56 | **0.58** | 25.73 | **26.32** |

achieving lower error rates while preserving high visual quality. This is confirmed by visual quality metrics such as BRISQUE, SSIM, and PSNR, which remain comparable to those of the original images. An intriguing future direction is to extend this setup to handle perturbations beyond Gaussian noise. One approach could involve pretraining the LISO steganographic encoder-decoder pair under such conditions before applying our framework. Alternatively, our method could be applied to steganographic frameworks other than LISO, particularly those explicitly designed to handle image perturbations, such as Tancik et al. [2020] and Bui et al. [2023].

