# OpenReview forum: "Neural Cover Selection for Image Steganography"
_NeurIPS.cc/2024/Conference — NeurIPS 2024 poster_

### Official Review · Reviewer_DaKE · 2024-07-05

**Soundness:** 3
**Presentation:** 2
**Contribution:** 2
**Rating:** 6
**Confidence:** 5

**Summary:**

This paper presents a steganographic cover optimization framework that can be used to enhance existing steganographic methods. The authors use a pre-trained DDIM to reconstruct the cover image, optimizing the latent in the process and thus reducing the message extraction error. Meanwhile, the authors deeply analyze the working principle of steganographic encoder and verify it experimentally and theoretically. The experimental results demonstrate that the method proposed in this paper reduces the message extraction error rate while improving the quality of stego images.

**Strengths:**

(1) From the perspective of steganographic cover optimization, the authors propose to use DDIM to adjust the cover image, which not only reduces the message extraction error rate, but also improves the quality of the stego image. Different from previous cover selection methods, the cover optimization method proposed in this paper has stronger interpretability.

(2) The authors deeply analyze the mechanism of steganographic encoder and conclude that "the encoder prefers to encode messages at pixel positions with smaller variance", which is verified by "waterfilling problem for Gaussian channels".

**Weaknesses:**

(1) The method framework description is not detailed enough. This paper implements the training of the whole steganography method by optimizing the message extraction loss. However, the framework in the paper contains two models: the pre-trained DDIM and the pre-trained LISO. Readers will naturally have questions: which of these two models are fixed? And which are trainable? Based on the contextual reading, I presume that the DDIM model parameters are updatable, while the LISO model is fixed. The authors should elaborate on these settings in Section 3.1.

(2) The experiment on steganalysis is not comprehensive enough. In Section 5.3, the authors evaluate the performance of the proposed steganography method to resist steganalysis. However, only the experiment of resisting the steganalysis network XuNet is designed, and the experiment of resisting the more advanced steganalysis network SRNet should be added to make the experimental results more convincing. Also, Section 5.3 lacks a detailed description of the experimental setup.

**Questions:**

(1) Unifying the term robustness, resistance to steganalysis should not be called robustness. In the field of steganography, robustness usually refers to the ability to resist channel perturbations, such as resistance to JPEG compression, resistance to Gaussian noise, and so on. The ability to resist steganalysis is often called security. For the definition of robustness, you can refer to the following articles:

1. Zeng, K., Chen, K., Zhang, W., Wang, Y., & Yu, N. (2023). Robust steganography for high quality images. IEEE Transactions on Circuits and Systems for Video Technology, 33(9), 4893-4906.
2. Zeng, K., Chen, K., Zhang, J., Zhang, W., & Yu, N. (2024). Towards Secure and Robust Steganography for Black-box Generated Images. IEEE Transactions on Information Forensics and Security.

(2) In Section 5.3, the settings of steganalysis experiments should be given, e.g., how many samples are used for training and how many samples are used for testing. In addition, experiments for resisting the more advanced steganalyzer SRNet should be added.

(3) In section 3.1 detail how to implement the description of latent optimization, is the message extraction loss ‖m-m ̂‖ used to update the pre-trained DDIM?

(4) The work in this paper is to invert the cover image to make it more suitable for steganography. Since this process does not involve "selection", I think it is not appropriate to call it "cover selection", you can consider modifying it to "cover generation" or "cover reconstruction".

(5) In Section 5.3, the performance index for evaluating resistance to XuNet is "detection rate", and the detailed definition of "detection rate" should be given here. Meanwhile, to evaluate the performance of resisting steganalysis, we usually use the index of "error rate", which is defined as error rate = N_false/N_test, where N_false is the number of classification errors and N_test is the total number of samples. The closer the error rate is to 50%, the better the performance is.

**Limitations:**

The authors have addressed the limitations.

---

> ### Author Rebuttal · Authors · 2024-08-07
>
> We appreciate the reviewer's recognition of our contributions. Our work advances steganographic cover optimization by using DDIM to adjust the cover image, reducing extraction errors and improving stego image quality. Unlike previous methods, our approach offers stronger interpretability. We show the encoder prefers low-variance pixels for message encoding, verified by the waterfilling algorithm for Gaussian channels.
>
> We address the two weaknesses as follows.
>
> 1. **Clarification on fixed vs. trainable components**: The DDIM and LISO models are pre-trained and remain fixed throughout the process. The latent vector $x_T$  generated by the DDIM's forward pass is the only component optimized by minimizing the extraction loss $||m-\hat{m}||$ (lines 141-142). This method is both practical and efficient, as it avoids the need for costly fine-tuning of DDIM. We will clarify this point in the revised manuscript.
>
> 2. **Additional experiments using SRNet**: We conducted experiments using SRNet, and the results are presented in Table 1 of the supplementary material. Our observations indicate that the images generated by our framework effectively resist steganalysis by SRNet. This is evidenced by the significant drop in detection rate when transitioning from scenario 1 to scenario 2. As a reminder, in scenario 2, we exploit the differentiability of the steganalyzer (SRNet) and incorporate an additional loss term to account for steganalysis . We will ensure this information is clearly presented in the manuscript.
>
> Below, we provide a detailed, line-by-line response to each of the questions.
>
> 1. **Security and robustness terminology**: Indeed; it makes more sense to use "security" for resistance to steganalysis and "robustness" for resistance to channel perturbations. We will update the terminology in the manuscript accordingly.
>
> 2.  **Steganalysis settings**: Regarding the settings of the steganalysis experiments, we adhered to the default values specified in the LISO paper. We will add detailed information about the number of samples used for training and testing in the revised manuscript.
>
> 3. **Latent optimization description**: Thank you for your suggestion, we will make this process clearer in the text. The DDIM is pre-trained and fixed during the entire process, as is the LISO model. The latent vector $x_T$ ​ produced by the DDIM’s forward pass is the only entity being optimized by minimizing the extraction loss $||m-\hat{m}||$ (lines 141-142). This approach is practical and efficient, as it avoids the costly fine-tuning of the DDIM.
>
> 4.  **Reconsidering terminology, replacing “Cover Selection”**: Thanks for the suggestion. We are considering several options such as "cover reconstruction," "cover fine-tuning," or "cover re-generation" to better describe our process.
>
>
>
> 5. **Clarification on detection and error rates in steganalysis**: Thank you for your feedback. We define the detection rate as detection rate=N_true/N_test. We will include this definition in the revised manuscript. Our use of this definition follows the steganographic framework presented in LISO [1]. We will also clarify its connection to the "error rate" as you described, noting that the closer the error rate is to 50%, the better the performance in resisting steganalysis. We will ensure these definitions and their implications are clearly explained.
>
>    [1] Xiangyu Chen, Varsha Kishore, and Kilian Q Weinberger. “Learning iterative neural optimizers for image steganography.” In International Conference on Learning Representations, 2023.

---

> > ### Comment · Reviewer_DaKE · 2024-08-12
> > **Comment on Authors' Response**
> >
> > The authors have addressed most of my concerns, so I will raise my score.

---

> ### Author Response · Authors · 2024-08-13
>
> Thank you for engaging in our discussion. We genuinely appreciate your constructive feedback.

---

### Official Review · Reviewer_XM9B · 2024-07-11

**Soundness:** 3
**Presentation:** 3
**Contribution:** 3
**Rating:** 4
**Confidence:** 4

**Summary:**

This paper introduces an innovative cover selection framework that optimizes within the latent space of pretrained generative models to identify the most suitable cover images, distinguishing it from traditional exhaustive search methods. This approach offers significant advantages in both message recovery and image quality. Furthermore, the paper presents an information-theoretic analysis of the generated cover images, revealing that message hiding predominantly occurs in low-variance pixels, consistent with the waterfilling algorithm principles in parallel Gaussian channels. Extensive experiments validate the superior performance of this framework.

**Strengths:**

1.This paper describes the limitations of current cover selection methods and introduces a novel, optimization-driven framework that combines pretrained generative models with steganographic encoder-decoder pairs.
2.The results demonstrate that the error rates of the optimized images are an order of magnitude lower than those of the original images under specific conditions. Impressively, this optimization not only reduces error rates but also enhances the overall image quality, as evidenced by established visual quality metrics.
3.This paper investigates the workings of the neural encoder and finds it hides messages within low variance pixels, akin to the water-filling algorithm in parallel Gaussian channels. In addition, the authors observe that this selection framework increases these low variance spots, thus improving message concealment.

**Weaknesses:**

We observed that the quality of the images used in the experimental section of the paper was generally not high, whether original or algorithmically generated. This may be due to the limitations of the dataset itself. Therefore, future experiments with high-quality images are necessary to further verify the effectiveness of this framework.

**Questions:**

1.You mentioned a JPEG layer in section 5.2, and I suggest that you add a brief description of this JPEG layer in your paper.
2.In Figure 11 of Appendix F, I observe that the subgraphs in the third row and third column are fuzzier than the other subgraphs in the third row. What do you think is the reason for this?
3.You selected the BigGAN network and DDIM network (Kim et al. 2022) in the experimental part of the paper. What are the advantages of these two networks?
4.What is the role of weight initialization in Figure 2?
5.In the introduction, I did not find this article in the references (Evsution et al. 2018), please check your references carefully.

**Limitations:**

The author has proposed an improved algorithm to address the limitations of current steganography methods, which has partially alleviated these constraints. It is recommended that in future work, the author focuses on enhancing image quality without compromising the algorithm's security performance.

---

> ### Author Rebuttal · Authors · 2024-08-07
>
> We appreciate the reviewer's recognition of our work, including (a) identifying limitations of existing cover selection methods, (b) integrating pretrained generative models with steganographic encoder-decoder pairs, and (c) demonstrating that our neural encoder hides messages within low variance pixels, similar to the water-filling algorithm. Our approach yields lower error rates and enhanced image quality, confirmed by visual metrics, and increases low variance pixels, further improving message concealment.
>
> We would like to provide pointers to the experiments and key results in the manuscript and appendix that we strongly believe address the main weaknesses and limitations identified in the review.
>
> 1. **Experiments with high-quality images:** We agree that this is an important aspect and would like to emphasize that we conducted extensive experiments using high-quality images from the CelebA-HQ dataset at 1024x1024 resolution and Animal Faces-HQ (AFHQ) dataset at 512x512 resolution. The qualitative results can be found in Fig. 10 of the appendix, with additional results shown in Fig. 1 and Fig. 2 of the supplementary material. These figures illustrate the high quality of the CelebA-HQ and AFHQ images. The effectiveness of our framework is further validated by the metrics presented in Table 2 of the main text, including Error Rate, BRISQUE, PSNR, and SSIM. These results indicate that we achieve similar improvements compared to the baselines for both high-quality and low-quality images.
>
> 2. **Enhancing image quality without compromising the algorithm's security performance**: We would like to clarify that the algorithm's security performance is not compromised. As shown in Table 4 of the manuscript, XuNet's detection rate using the AFHQ dataset (high-quality 512x512 images) remains similar, if not lower, before and after applying our framework, indicating that our approach maintains security performance. Additionally, the high quality of our AFHQ images is validated in Table 2, measured by metrics such as BRISQUE, SSIM, and PSNR.
>
> Perhaps these setups were not clear, and we will clarify them in the revised manuscript. We kindly ask the reviewer to consider these results in their review.
>
> We provide answers to the questions and suggestions below.
>
> 1. **JPEG layer description:** Thanks for the suggestion. We agree that a brief description of the JPEG layer in Section 5.2 will enhance the clarity of our paper. We will include this in the revised manuscript.
>
> 2. **Fig. 11 third row third column fuzziness:** The subgraph in the third row and third column does appear fuzzier than the others. This issue is an outlier and has not been observed in CelebA-HQ, AFHQ, or other ImageNet classes. It arises due to the LISO framework, on which our method is built. LISO is the state-of-the-art steganographic encoder-decoder, which is why we chose to build our method on it. However, training LISO on ImageNet is challenging because of the vast number of classes. To address this, we fine-tuned LISO using additional owl images obtained through data augmentation techniques such as rotation and flipping, and achieved a slightly better-quality image, as shown in Fig. 3 of the supplementary material.
>
>    We would also like to emphasize that the main focus of this work is to design a cover selection algorithm, not the steganographic encoder and decoder. While we have made improvements to the owl image, further investigation into enhancing LISO’s performance on ImageNet would be necessary for even better results, which was beyond the scope of the main paper.
>
> 3. **Advantage of BigGAN network and DDIM network (Kim et al. 2022):**  BigGAN and the diffusion model from Kim et al. (2022) are among the state-of-the-art generative models, known for their high-quality image generation capabilities. Both models are open-sourced, making them accessible for replication. Additionally, the papers introducing these models are highly cited, indicating their significant impact and validation within the research community. We would also like to note that our approach is applicable to other generative models as well.
>
> 4. **Clarification on the role of weight initialization in Fig. 2:** As described in Section 3.1 of the manuscript, our process consists of two steps. In step 1, the initial cover image goes through the forward diffusion process to get the latent $x_T$. This serves as the initialization for the next step. In step 2, we optimize the already initialized elements of $x_T$ such that when it goes through the backward diffusion process, the output image minimizes the message reconstruction loss. In other words, we treat $x_T$ as a learnable weight matrix initialized from step 1, and its weights are updated via step 2 using gradient descent. We hope this addresses your question, and we will add more details to this section to make it clearer.
>
> 5. **Reference pointer (Evsutin et al. 2018):** We have carefully checked our references, and the article by Evsutin et al. (2018) is included (lines 350-352). If there are any specific concerns or if you need further clarification, please let us know.

---

### Official Review · Reviewer_kQa7 · 2024-07-13

**Soundness:** 3
**Presentation:** 3
**Contribution:** 3
**Rating:** 5
**Confidence:** 4

**Summary:**

This paper presents a novel framework for cover selection in image steganography to enhance the message recovery performance, which optimizes the latent code. The effectiveness of this approach is validated through intensive experiments. Additionally, the paper empirically analyzes intriguing behaviors occurring during the message hiding process inside the message encoder.

**Strengths:**

* Introducing a novel approach for cover selection in image steganography aimed at minimizing error rates.
* This straightforward approach optimizes the latent code by minimizing message loss, yet yields promising results.
* The proposed method is thoroughly validated, demonstrating strong performance across various metrics including message recovery, robustness, and security (steganalysis).

**Weaknesses:**

* The optimization of latent variables reduces recovery errors but alters the original image content (Fig. 1), making it impractical for real-world use. Additionally, as depicted in Figure 10 (Appendix E), steganographic images may exhibit unnatural visual characteristics, potentially leading to easy detection.
* While the inclusion of an additional loss term for steganalysis is pivotal for setting 2, the author fails to address this point. Moreover, the results in setting 1 are puzzling: why does detection increase with 1bpp but not with 2bpp, despite the increased payload?

**Questions:**

* Why the authors do not compare with others steganography frameworks such as StegaStamp [1], RoSteALS [2], UDH [3], or StegaStyleGAN [4]
* Can the authors show more qualitative results between the original and encoded images? Is there any effect of the optimization on the orignal images?
* Can the authors explain why the percentages of the identified high-message positions are encoded in low-variance pixels will affect the flexibility for data embedding ?

[1] Tancik, Matthew, Ben Mildenhall, and Ren Ng. "Stegastamp: Invisible hyperlinks in physical photographs." Proceedings of the IEEE/CVF conference on computer vision and pattern recognition. 2020. \
[2] Bui, Tu, et al. "Rosteals: Robust steganography using autoencoder latent space." Proceedings of the IEEE/CVF Conference on Computer Vision and Pattern Recognition. 2023. \
[3] Zhang, Chaoning, et al. "Udh: Universal deep hiding for steganography, watermarking, and light field messaging." Advances in Neural Information Processing Systems 33 (2020): 10223-10234. \
[4] Su, Wenkang, Jiangqun Ni, and Yiyan Sun. "StegaStyleGAN: Towards Generic and Practical Generative Image Steganography." Proceedings of the AAAI Conference on Artificial Intelligence. Vol. 38. No. 1. 2024.

**Limitations:**

The authors adequately addressed the limitations, and broader impact in this paper.

---

> ### Author Rebuttal · Authors · 2024-08-07
>
> We express our gratitude to the reviewer for their valuable feedback, and we will address each point in detail.
>
> **Weaknesses**:
>
> - **Image alteration:** We acknowledge the concern regarding the alteration of the original image content due to the optimization of latent variables. However, it is important to highlight that such modifications are not only common but also well-founded in the literature.
>
>   **a) Established practice in research:** Our approach of altering the cover image is in line with established practices in the field of cover selection. For example, studies such as [1] and [2] analyzed different cover images and selected the most suitable ones to improve the robustness and accuracy of message extraction. These works provide concrete evidence and justification for the alteration of cover images, demonstrating that such modifications are beneficial for enhancing performance in steganographic applications.
>
>     [1] Farzin Yaghmaee and Mansour Jamzad. Estimating watermarking capacity in gray scale images based on image complexity.  EURASIP Journal on Advances in Signal Processing, 2010:1–9, 2010.
>
>     [2] Zichi Wang and Xinpeng Zhang. Secure cover selection for steganography. IEEE Access, 7:57857– 57867, 2019.
>
>   **b) Preservation of image semantics:** While our algorithm alters the cover image to minimize decoding error, it does not change its meaning or semantics. In semantic communications, these modifications are acceptable as the focus is on accurately transmitting essential information, not preserving the exact original image content.
>
> - **Unnatural visual characteristics (Fig. 10), potentially leading to easy detection**: We want to clarify that the algorithm's detection accuracy remains intact. Table 4 of the manuscript demonstrates that XuNet's detection rate with the AFHQ dataset stays consistent or even decreases after applying our framework, showing that our method does not compromise security performance.
>
> - **Loss term for steganalysis:** Our paper addresses the inclusion of an additional loss term for steganalysis, as outlined in Section 5.3 (lines 304-306). We incorporate the steganalyzer’s logit output into the loss function. Our results in Table 4 show that, when embedding 4 bits per pixel, detection accuracy drops from 97.35% to 8.6% after incorporating the loss, indicating improved resistance to steganalysis. We will clarify this further in the main text.
>
> - **Unexpected decline in detection rate:** Indeed, it is puzzling that detection decreases despite the increased payload. This effect is not related to our introduced method but is inherent to the LISO framework on which our method is based. As shown in Table 5 of the LISO manuscript [1], they also observe this unexpected behavior where the detection rates do not increase consistently with higher payloads. This phenomenon is not fully understood.
>
>   We chose to use LISO because it is a state-of-the-art steganographic framework. However, we recognize that this reliance on LISO also means inheriting some of its unexplained behaviors. Future work could explore approaches combining LISO with other frameworks or develop novel techniques to address detection rate inconsistencies with different payloads. We will highlight this in our paper and acknowledge the need for further research to understand and improve the framework.
>
>   [1] Xiangyu Chen, Varsha Kishore, and Kilian Q Weinberger. “Learning iterative neural optimizers for image steganography.” In International Conference on Learning Representations, 2023.
>
> **Questions:**
>
> - **Comparisons to other schemes:** While related, the mentioned frameworks are designed for different settings than those we used. The approaches in [1, 2] involve networks that learn algorithms **robust to image perturbations** between the encoder and decoder, whereas our settings assume **no such perturbations**. In [3], the focus is on hiding secret **images** within cover images, unlike our method, which hides secret **binary messages**. Lastly, [4] deals with coverless steganography, mapping secret messages to latent vectors for a generative adversarial network, which then generates **random cover images**. In contrast, our settings involve embedding secret messages into **given cover images** using a dedicated encoder.
>
> - **Additional qualitative results:** We have included additional qualitative results comparing the original and encoded images in Fig. 1 of the supplementary material.
>
> - **Effect on optimized images:**
>   1. **Relationship with Image Complexity Metrics:** We would like to refer the reviewer to Fig. 12 of Appendix G, where we conducted extensive simulations to answer this important question. In our experiments on the AFHQ dataset, we analyzed the relationship between the decoding error and various image complexity metrics: entropy, edge density, compression ratio, and color diversity. Our results show a negative correlation between the decoding error and the first three metrics, and a positive correlation with color diversity. We conjecture that optimizing for decoding error is affecting these image complexity metrics. While these findings are encouraging, more research is needed to fully understand the effects of optimization on the original images.
>
>   2. **Increase in Low-Variance Pixels:** As discussed in Section 4.3 of the main text, the optimization process increases the number of low-variance pixels in the images from 81.6% to 92.4%. This is related to our discussion about the algorithm hiding data in low-variance spots. When the number of low-variance pixels increases, the encoder has more flexibility and options for data hiding, resulting in better performance.
>
>    We will include more details of this discussion in the manuscript to highlight the need for further investigation.
>
> - **Clarification, encoding in low-variance pixels:** We would like to refer the reviewer to part (3) of the global response. We hope this description addresses the question.

---

> > ### Comment · Reviewer_kQa7 · 2024-08-12
> >
> > Thanks for the authors's detailed response. The authors address well my concerns about the framework, however, the authors should include more discussion about several issues raised in the revised version. The authors's assumption which is no pertubation in images may not be practical in the secret communication, I suggest the authors should consider this scenario and provide the comparative results with several methods such as RoSteALS, or StegaStamp. I would like to keep my original rating.

---

> ### Author Response · Authors · 2024-08-13
> **Response to Reviewer kQa7's Comment**
>
> We thank the reviewer for the response.
>
> We would like to clarify three key points:
>
> 1) The primary focus of our work is on developing a cover selection algorithm rather than a complete steganographic encoding-decoding framework. Our method is versatile and can be integrated with various steganographic frameworks, including RoSteALS and StegaStamp. The assumption of no perturbation aligns with the steganographic frameworks described in [1,2,3]. Among those 3, we specifically selected the LISO framework [3] due to its status as the state-of-the-art. This line of work operates without assuming any perturbations, which is why perturbation results are not presented in our work.
>
>    [1] Kevin Alex Zhang, Alfredo Cuesta-Infante, Lei Xu, and Kalyan Veeramachaneni. “Steganogan: High capacity image steganography with gans.” arXiv preprint arXiv:1901.03892, 2019.
>
>    [2] Varsha Kishore, Xiangyu Chen, Yan Wang, Boyi Li, and Kilian Q Weinberger. “Fixed neural network steganography: Train the images, not the network.” In International Conference on Learning Representations, 2021.
>
>    [3] Xiangyu Chen, Varsha Kishore, and Kilian Q Weinberger. “Learning iterative neural optimizers for image steganography.” In International Conference on Learning Representations, 2023.
>
> 2) Although we do not provide perturbation results, we present results related to JPEG distortion (Section 5.2), which is highly relevant to real-world applications and offers a meaningful representation of our method’s effectiveness.
>
> 3) Regarding the request to “provide comparative results with several methods such as RoSteALS or StegaStamp”, we are uncertain about the exact nature of the comparative results the reviewer is seeking. As mentioned earlier, our focus is on designing a cover selection algorithm for a given steganographic encoder-decoder pair. If RoSteALS outperforms StegaStamp, then our cover selection framework applied to RoSteALS would also outperform its application to StegaStamp. If the reviewer is suggesting a comparison of our framework when applied to LISO versus RoSteALS or StegaStamp, it would not be a fair comparison, as LISO is trained under different conditions than the other methods.

---

> > ### Comment · Reviewer_kQa7 · 2024-08-14
> >
> > Thank you for your answer. I'm aware that RoSteALS, StegaStamp, and LISO are under different training setting. However, given that response, there are no quantitative results to show that your method is robust with other steganographic frameworks including RoSteALS and StegaStamp. Moreover, in steganography, we do care about the distortion scenarios, hence, even though your report the results related to JPEG compression I think it would not be enough. I would like to suggest the authors can improve these concerns in the future work. Again, I would like to keep my rating.
> >
> > Thanks for your work!

---

### Author Rebuttal · Authors · 2024-08-07

1. We thank all the reviewers for acknowledging our contributions and providing constructive feedback. Our ***key contributions***  include (a) identifying limitations of existing cover selection methods, (b) integrating pretrained generative models with steganographic encoder-decoder pairs for cover fine-tuning to minimize the error extraction loss, and (c) conducting theoretical studies to shed light on the inner workings of learning-driven steganography methods, drawing analogies to the water-filling algorithm in signal processing.

2. We conducted ***additional experiments*** to address the reviewers' concerns, including: (1) steganalysis results on SRNet (requested by Reviewer DaKE), (2) additional qualitative comparisons between the original and encoded images (requested by Reviewer kQa7), and (3) an experiment to enhance image quality, addressing the concern raised by Reviewer XM9B. The results to these experiments are included in the ***attached PDF***.

3. We also clarify an important question raised by Reviewer kQa7 regarding our analysis of the inner workings of the steganographic encoder and its analogy to the water-filling algorithm:

   - ***Analogy:*** We view the process of hiding secret messages as sending information through N parallel communication channels, where N represents the number of pixels in an image. In this analogy:

     - Each pixel functions as an individual communication link.
     - The secret message to be hidden and eventually recovered acts as the signal.
     - The cover image, which hosts the hidden message, introduces noise that is unknown to the decoder.

   - ***Advantage of low variance pixels in hiding messages:*** The analogy above immediately helps understand the role of pixel variances within a cover image in hiding messages. Intuitively, low-variance pixels within a cover image indicate low noise, reducing uncertainty for the decoder and enhancing message recovery.

   - ***Key Observation. High-message positions are highly aligned with low-variance pixels:*** In Section 4.1, we make an observation that the high-message positions are highly aligned with low-variance pixels, meaning the steganographic encoder actively utilizes the low-variance pixel positions for hiding the messages, which is a highly desired and natural behavior. Our findings show that **81.6%** of high-message regions align with low-variance pixels, leveraging lower noise to enhance recovery accuracy. **We highlight that we are the first to make this observation, despite there being several relevant works on learning-driven steganography; none of these prior studies conducted an interpretation analysis of the encoder to uncover this behavior, as also acknowledged by Reviewers XM9B and DaKE.**

   - ***Theoretical Reasoning behind our Key Observation:*** Interestingly, we find that the learned message embedding behavior closely aligns with the waterfilling strategy, the theoretically optimal embedding strategy for parallel Additive Gaussian Noise channels. This strategy involves embedding more messages in lower-variance pixel positions, which increases message recovery accuracy. Surprisingly, steganography methods tend to adopt this strategy implicitly, without explicit training to do so, as discussed in Section 4.2. To the best of our knowledge, this is the **first instance where theoretical studies are used to shed light on the inner workings of learning-driven steganography methods.**



   - ***How is this discussion relevant to our cover selection/fine-tuning problem?*** In Section 4.3, we observe a significant increase in the number of low-variance pixels within a cover image after applying our framework. Our findings demonstrate that **92.4%** of high-message regions now align with low-variance pixels, compared to **81.6%** before the optimization of the cover image. This indicates a significant enhancement in the encoder's performance, as it efficiently leverages the low-variance pixel positions to conceal the messages.

---

### Decision · Program_Chairs · 2024-09-25

**Decision:**

Accept (poster)

**Comment:**

This paper was reviewed by three experts in the field and got mixed recommendations. Both reviewers kQa7 and DaKE initially rated Borderline Accept. The authors addressed most of their concerns in the rebuttal and discussion, except for missing quantitative evidence proving the generalization of the proposed algorithm to work with other steganography techniques. Hence, reviewer DaKE raised his score to Weak  Accept, while reviewer kQa7 kept his core unchanged. Reviewer XM9B  initially rated Borderline Reject, and he did not provide any update after the rebuttal. The ACs checked and found the rebuttal well-addressed the reviewer's concerns.

Based on the observation above, the decision is to recommend the paper for acceptance to NeurIPS 2024. The authors should include the discussions and experimental results from the rebuttal in the camera-ready version. Some notable suggested modifications include clearer writing (Reviewer DaKE), three additional qualitative and quantitative, clarification on the analysis of the inner workings of the steganographic encoder and its analogy to the water-filling algorithm (Reviewer kQa7), and discussion on unexpected outcomes related to the LISO framework (Reviewers kQa7 and XM9B). The authors are also encouraged to add an extra experiment showing the robustness of the proposed method when combined with other steganographic frameworks like RoSteALS and StegaStamp.

We congratulate the authors on the acceptance of their paper!